# The adhesion function of the sodium channel beta subunit (β1) contributes to cardiac action potential propagation

Rengasayee Veeraraghavan[1,2]*, Gregory S Hoeker[1,2], Anita Alvarez-Laviada[3], Daniel Hoagland[1,2], Xiaoping Wan[4], D Ryan King[1,2,5], Jose Sanchez-Alonso[3], Chunling Chen[6], Jane Jourdan[1,2], Lori L Isom[6], Isabelle Deschenes[4,7], James W Smyth[1,2,8], Julia Gorelik[3], Steven Poelzing[1,2,9], Robert G Gourdie[1,2,9]*

[1]Virginia Tech Carilion Research Institute, Virginia Polytechnic University, Roanoke, United States; [2]School of Medicine, Virginia Polytechnic University, Roanoke, United States; [3]Department of Myocardial Function, Imperial College London, London, United Kingdom; [4]Heart and Vascular Research Center, MetroHealth Medical Center, Department of Medicine, Case Western Reserve University, Cleveland, United States; [5]Graduate Program in Translational Biology, Medicine and Health, Virginia Tech, Virginia, United States; [6]Department of Pharmacology, University of Michigan Medical School, Ann Arbor, United States; [7]Department of Physiology and Biophysics, Case Western Reserve University, Cleveland, Unites States; [8]Department of Biological Sciences, College of Science, Blacksburg, United States; [9]Department of Biomedical Engineering and Mechanics, Virginia Polytechnic University, Blacksburg, United States

*For correspondence:
saiv@vt.edu (RV);
gourdier@vtc.vt.edu (RGG)

Competing interests: The authors declare that no competing interests exist.

**Abstract** Computational modeling indicates that cardiac conduction may involve ephaptic coupling – intercellular communication involving electrochemical signaling across narrow extracellular clefts between cardiomyocytes. We hypothesized that β1(SCN1B) –mediated adhesion scaffolds *trans*-activating $Na_V$1.5 (SCN5A) channels within narrow (<30 nm) perinexal clefts adjacent to gap junctions (GJs), facilitating ephaptic coupling. Super-resolution imaging indicated preferential β1 localization at the perinexus, where it co-locates with $Na_V$1.5. Smart patch clamp (SPC) indicated greater sodium current density ($I_{Na}$) at perinexi, relative to non-junctional sites. A novel, rationally designed peptide, βadp1, potently and selectively inhibited β1-mediated adhesion, in electric cell-substrate impedance sensing studies. βadp1 significantly widened perinexi in guinea pig ventricles, and selectively reduced perinexal $I_{Na}$, but not whole cell $I_{Na}$, in myocyte monolayers. In optical mapping studies, βadp1 precipitated arrhythmogenic conduction slowing. In summary, β1-mediated adhesion at the perinexus facilitates action potential propagation between cardiomyocytes, and may represent a novel target for anti-arrhythmic therapies.

## Introduction

The concept that the heart beats as a functional syncytium composed of discrete muscle cells electrically coupled by gap junctions (GJs) is a foundational theory of modern cardiology (*Kléber and Rudy, 2004*). Moreover, as GJ remodeling occurs in multiple cardiac pathologies (*Jongsma and Wilders, 2000*; *Stroemlund et al., 2015*), it is widely held that disrupted GJ coupling is mechanistically central to arrhythmogenesis in heart disease (*De Vuyst et al., 2011*; *Palatinus et al., 2012*). Whilst a GJ-based paradigm for cardiac electrical coupling has been in place for over 50 years, a small group of mathematical biologists has proposed that cardiac conduction in health and disease may involve

ephaptic mechanisms (*Lin and Keener, 2010*; *Mann and Sperelakis, 1979*; *Mori et al., 2008*; *Pertsov and Medvinskiĭ, 1979*). Ephaptic conduction is conceived as involving the intercellular transmission of action potentials via ion accumulation/depletion transients occurring within narrow extracellular clefts between closely apposed myocytes. Further interest in this hypothesis has been stoked by provocative findings such as conduction still occurring in mice with knockout of the principal ventricular GJ protein, connexin43 (*Gja1*/Cx43) (*Gutstein et al., 2001*) and in humans with dominant negative mutations of GJA1, the gene encoding Cx43 (*Shibayama et al., 2005*).

Although the ephaptic hypothesis has remained controversial, owing to a lack of direct experimental evidence, theoretical studies have identified two key elements of a hypothetical structural unit that might support this alternative mechanism of conduction (*Mori et al., 2008*; *Hichri et al., 2018*; *Kucera et al., 2002*; *Lin and Keener, 2013*): 1) Close proximity (<30 nm) between the membranes of adjacent myocytes and 2) a high density of sodium channels at such points of close intermembrane apposition. These prerequisites for the formation of a cardiac ephapse pointed to its likely location in the intercalated disk (ID) where cardiac sodium channels are concentrated (*Maier et al., 2004*; *Petitprez et al., 2011*). Here, we report that α (*Scn5a*/Na$_V$1.5) and β (*Scn1b*/β1) subunits of cardiac sodium channels preferentially localize at the perinexus, a nanodomain at the edge of GJs, (*Rhett et al., 2011*) where the membranes of apposed cells routinely approach each other closely enough for ephapse formation (*Veeraraghavan et al., 2015*). Additionally, we demonstrate that adhesion mediated by the extracellular immunoglobulin (Ig) domain of β1 generates close proximity between the membranes of adjacent cells within perinexal nanodomains, likely enabling these to function as ephapses. Importantly, we identify dynamic changes in perinexal ultrastructure, secondary to compromised β1-mediated adhesion, as a novel mechanism for arrhythmogenic conduction defects.

## Results

In this study, we tested the hypothesis that Na$_V$1.5-rich nanodomains within the ID may enable ephaptic coupling between cardiac myocytes, and that the cell adhesion function of β1 may be important to this phenomenon. We present an array of experimental evidence detailing the nanoscale location, structural properties, and makeup of Na$_V$1.5-rich ID nanodomains and demonstrate that inhibition of β1-mediated adhesion disrupts these nanodomains, with proarrhythmic consequences.

### Sodium channel subunit proteins Nav1.5 and β1 differentially localize within ventricular ID nanodomains

The cardiac voltage-gated sodium channel is a trimer comprised of a pore-forming alpha subunit (Na$_V$1.5) and two non-pore-forming β subunits, the latter possessing an extracellular Ig domain, which facilitates adhesive interactions (*Namadurai et al., 2015*; *O'Malley and Isom, 2015*). To determine the distribution of the cardiac sodium channel complex we raised, characterized, and rigorously validated rabbit polyclonal antibodies against Na$_V$1.5 and β1. Both antibodies displayed intense immunolabeling of N-cadherin (N-cad) -positive IDs (*Figure 1A*, *Figure 1—figure supplement 1*) in laser scanning confocal microscope (LSCM) images of guinea pig (GP) ventricular tissue and single bands at the expected molecular weights in western blots (*Figure 1—figure supplement 1*; detailed results from validation studies are provided in supplementary material). Also, in dot blot experiments, both antibodies displayed selective affinity for their corresponding epitopes, with no evidence of cross-reactivity to each other's epitopes (*Figure 1—figure supplement 2*).

While the observation of ID-enrichment of Na$_V$1.5 (*Figure 1A*, *Figure 1—figure supplement 1A*) is consistent with previous studies (*Maier et al., 2004*; *Petitprez et al., 2011*), *en face* views of IDs reconstructed from LSCM optical sections revealed a previously unreported aspect of β1 organization (*Figure 1B*). In these end-on views of IDs, β1 displayed a lattice-like distribution with specific immunolocalization within N-cad-free interplicate sectors of the ID (*Figure 1B*, *Figure 1—figure supplement 3C*). Interplicate regions of the ID are well-characterized as being enriched in Cx43 GJs (*Severs, 2000*). In line with this, punctate Cx43 immunosignal corresponding to GJ demonstrated close association with the strands comprising the β1 lattice (*Figure 1—figure supplement 3B*).

The LSCM observations were confirmed by super-resolution STochastic Optical Reconstruction Microscopy (STORM) at sub-diffraction resolution (20 nm lateral, 40 nm axial) (*Figure 1C–F*). A

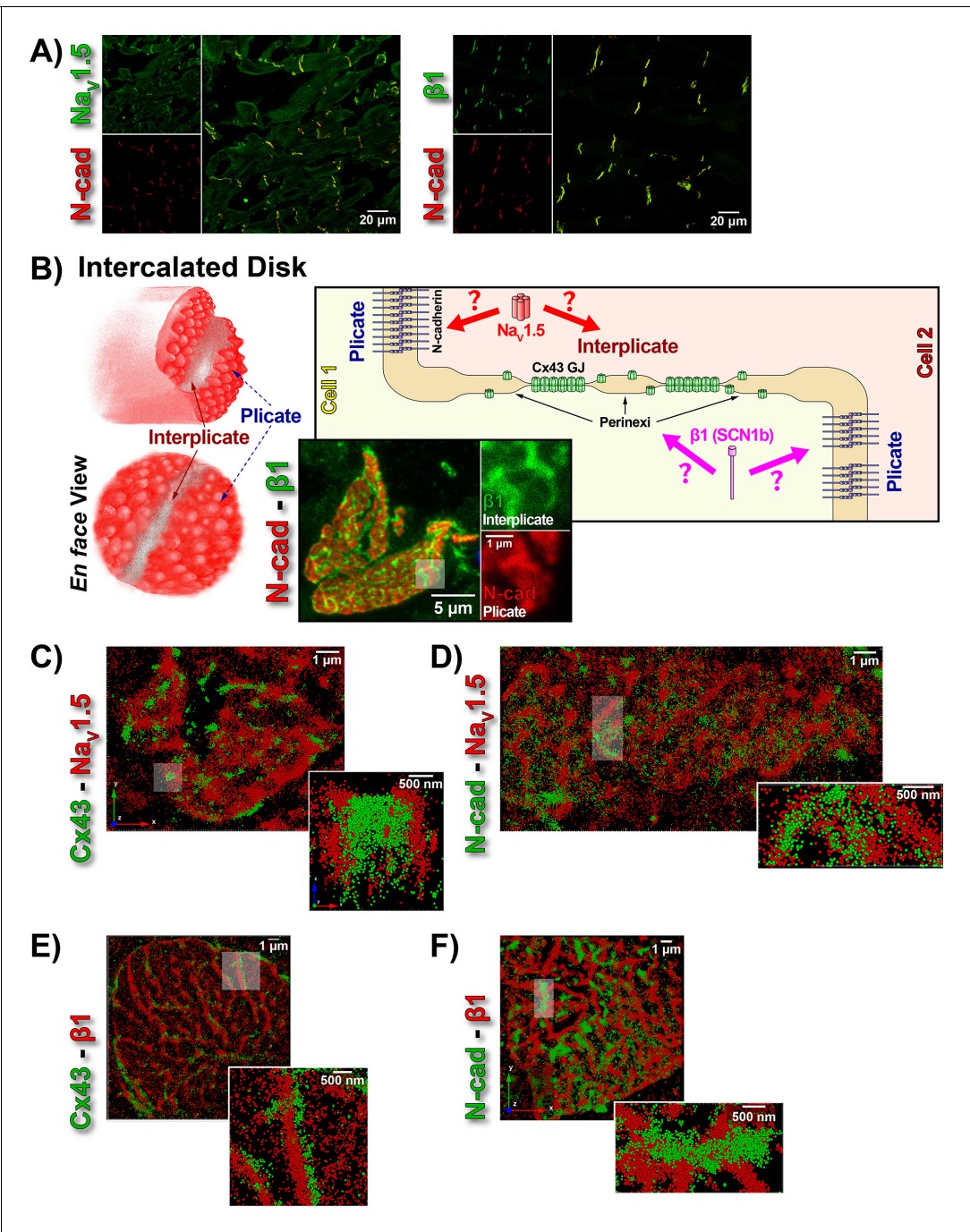

**Figure 1.** ID Localization of Na$_V$1.5 and β1. (A) Representative confocal images of GP left ventricular (LV) sections co-labeled for Na$_V$1.5 (green; top) / β1 (green; bottom) along with N-cad (red). (B) Schematic diagrams, and representative confocal images of N-cad (red), and β1 (green) illustrate the plicate, and interplicate regions of the ID when viewed *en face*. The cartoon on the right summarizes the essential question addressed using STORM: Where within the ID are Na$_V$1.5 and β1 localized? Representative STORM images showing x-y plane views of *en face* IDs labeled for: (C) Cx43 (green) and Nav1.5 (red), (D) N-cad (green) and Na$_V$1.5 (red), (E) Cx43 (green) and β1 (red), (F) N-cad (green) and β1 (red). Individual fluorophore molecules localized were at 20 nm lateral resolution, but are represented as 50 nm spheres to enhance visibility in print. The inset in C is shown rotated by 90° along the z-axis, and illustrates a Cx43 cluster flanked on either side by a cluster of Na$_V$1.5.

The online version of this article includes the following figure supplement(s) for figure 1:

**Figure supplement 1.** Validation of Na$_V$1.5 and β1 antibodies.
**Figure supplement 2.** Validation of Na$_V$1.5, β1 antibody specificity.
**Figure supplement 3.** ID localization of β1 via confocal immunofluorescence.

significant proportion of immunolocalized Na$_V$1.5 molecules were organized into clusters preferentially localized adjacent to clusters of Cx43 molecules (i.e. GJs; *Figure 1C*), while a population of Na$_V$1.5 molecules was also observed co-distributing with N-cad (*Figure 1D*). In contrast to Na$_V$1.5, β1 molecules organized into a lattice-like distribution with β1 strands punctuated by tight side-by-side association with Cx43 clusters (*Figure 1E*), but extending almost exclusively through N-cad-free interplicate zones of the ID (*Figure 1F*).

## STORM-RLA indicates Na$_V$1.5 distributes between two pools within the intercalated disk

To quantitatively assess the overlapping, but distinct distributions of β1 and Na$_V$1.5 relative to Cx43 (GJs) and N-cad (plicate zone/adherens junctions) within the ID, we used STORM-based relative localization analysis (STORM-RLA; *Figure 2*) (*Veeraraghavan and Gourdie, 2016*). Briefly, relative localization of co-labeled proteins is quantitatively assessed by detection of clusters of localized molecules, and measurement of overlap, and closest distances between clusters.

In accordance with visual assessment (*Figure 1*), STORM-RLA indicated that nearly half of ID-localized Na$_V$1.5 clusters (48.1%) were located adjacent Cx43 GJ (*Figure 2A,C*), a sub-population we term the *perinexal pool*. This pool included 31% of Na$_V$1.5 clusters that did not overlap Cx43 clusters, and an additional 17.1% which tangentially overlapped Cx43 clusters (*Figure 2A,B,C*). A second, *plicate pool* of Na$_V$1.5, localized to N-cad-rich plicate ID regions, was found to contain 29% of ID-localized Na$_V$1.5 (*Figure 2A,B,D*), consistent with previous reports from us (*Veeraraghavan and Gourdie, 2016*) and the Delmar group (*Leo-Macias et al., 2016*). Closely

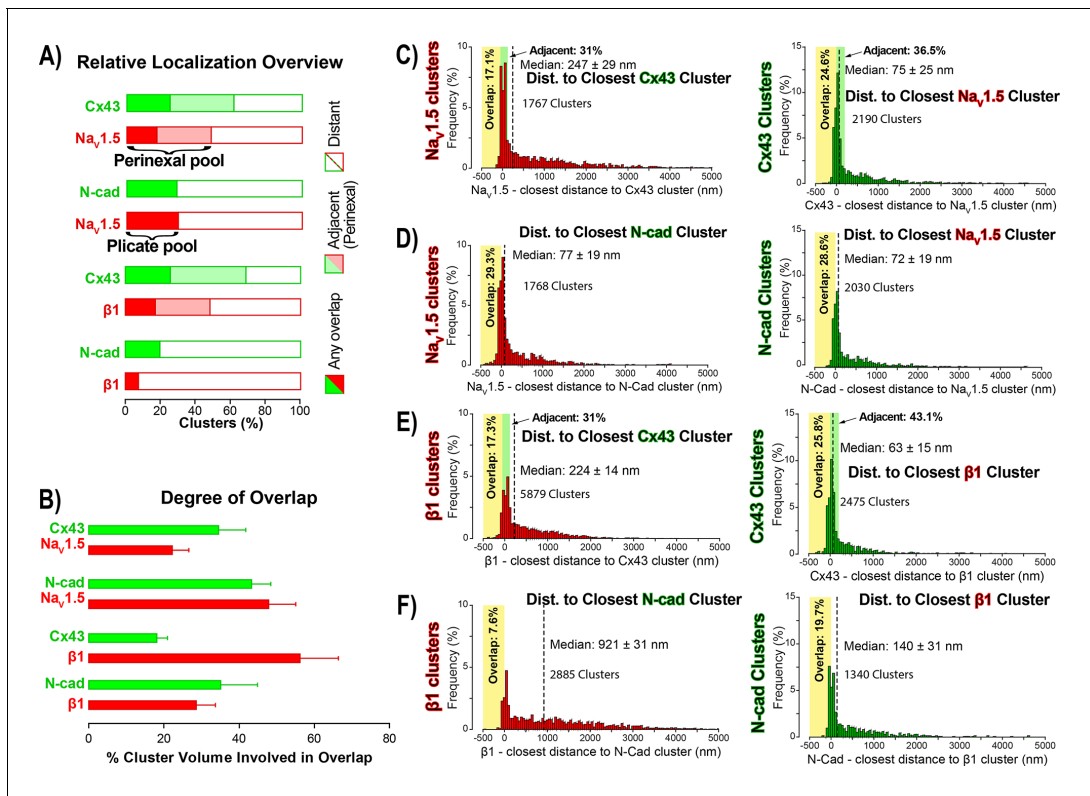

**Figure 2.** STORM-RLA quantification of Na$_V$1.5 and β1 localization. (**A**) A graph summarizing STORM-RLA analysis of relative localization between clusters of co-labeled proteins. The solid bars indicate clusters with any overlap, the shaded bars represent adjacent clusters (corresponding to perinexal localization), and the clear bars indicate clusters distant from each other. (**B**) A summary graph of the degree of overlap, that is the fraction of cluster volume involved in overlap for those clusters, which demonstrated any overlap (corresponding to the filled bars in **A**). (**D–F**) Summary histograms generated by STORM-RLA show the closest inter-cluster distances between clusters of co-labeled proteins (n = 3 hearts, four image volumes per heart). The yellow boxes on each plot highlight negative inter-cluster distances, which correspond to overlapping clusters. Dashed black lines mark the median values. The green boxes indicate overlap of Na$_V$1.5/β1 clusters with perinexal regions surrounding Cx43 clusters (extending 200 nm from the GJ/Cx43 cluster edge [*Veeraraghavan et al., 2015*]).

paralleling the Na$_V$1.5 case, nearly half (48.3%) of ID-localized β1 clusters were also identified within perinexal nanodomains (*Figure 2A,E*). However, only 7.6% β1 clusters were co-located with N-cad clusters (*Figure 2A,F*), suggesting that an overwhelming 92.4% of ID-localized β1 clusters are located in N-cad-free, GJ-rich interplicate ID regions. In summary, the pore-forming subunit Na$_V$1.5 is divided between two significant pools within the ID, a perinexal pool, and a plicate pool; however, the β1 subunit is mainly localized in perinexal, and other interplicate ID sites.

## βadp1 – a rationally designed inhibitor of β1-mediated adhesion

The extracellular Ig domain of β1 facilitates adhesive interactions, including *trans* homophillic interaction with β1 molecules on neighboring cells (*Calhoun and Isom, 2014*). In order to assess this intercellular adhesion function, as well as to create an assay model for identification of β1 adhesion antagonists, we quantified intercellular junctional resistance changes in cells heterologously overexpressing β1 (1610 β1OX) using electric cell-substrate impedance sensing (ECIS). This technique measures the electrical impedance offered by a monolayer of cells to current flow between electrodes located above, and below the monolayer, and previous studies demonstrate that the resistance of the extracellular cleft at cell-cell junctions (junctional resistance) is well-reflected by measurements at low frequencies (62.5–4000 Hz) (*Moy et al., 2000*; *Tiruppathi et al., 1992*). Junctional resistance was markedly higher (>3 fold) in 1610 β1OX cells, compared to cells without β1 (1610 Parental; *Figure 3A*), consistent with the expression of β1 increasing levels of adhesion between the cells. Immunosignals for tight junction proteins and Ca$^{2+}$-dependent cadherins were below detectable levels, and did not change with β1 expression (data not shown), suggesting that these proteins were unlikely to account for adhesion observed between 1610 cells.

Our strategy for preparing a rationally designed antagonist of β1-mediated adhesion was based on an approach successfully applied to two other adhesion proteins of the Ig domain family that share homologies with β1: N-cad (*Williams et al., 2000a*; *Williams et al., 2000b*) and desmoglein-1 (*Schlipp et al., 2014*). This method involves synthesizing a peptide mimicking the adhesion sequence within the Ig loop domain. The location of the adhesion receptor in sodium channel β Ig domain has been characterized in previous structure and function studies (*Brackenbury and Isom, 2011*; *Malhotra et al., 2000*; *Patino et al., 2009*). Based on the recently resolved crystal structure of β3, and structural data from cryo-electron microscopy studies of β1 in the electric eel, we generated a molecular homology model of β1 using SWISS-MODEL (*Figure 3B*) (*Namadurai et al., 2014*). We then identified an amino acid sequence (1FVKILRYENEVLQLEEDERF20: βadp1) within the β1 Ig domain homologous to the desmoglein-1 Ig loop adhesion sequence - a peptide mimetic of which acts as a competitive inhibitor of desmosomal adhesion (*Harrison et al., 2016*). To assess the interaction of βadp1 with β1, and thus its potential ability to act as a competitive inhibitor of β1 adhesion, we allowed the peptide to bind freely to the β1 homology model in silico, determining that it associated with the adhesion surface of the Ig loop (*Figure 3C*). MM-GBSA refined docking of βadp1 using Maestro displayed a low energy conformation (ΔG binding −44.47) at this location. The predicted binding pose revealed stabilization of this conformation via a network of intramolecular hydrogen bonds, and electrostatic interactions between the C-terminal end of βadp1 and polar residues within the β1 adhesion domain, as well as accommodation of hydrophobic residues of βadp1 within pockets along the β1 adhesion surface. Sequences of similar length from the Ig loop outside of the adhesion domain, as well as a randomized βadp1 sequence (βadp1-scr), did not show propensity to interact with the adhesion surface of the β1 extracellular domain. Substituting the basic arginine (R) at position 19 of βadp1 with acidic aspartic acid (D) yielded a sequence (βadp1-R85D) with significantly abrogated binding to the β1 model in silico (ΔG binding −18.05). The arginine at this position of βadp1 corresponds to R85 on full length β1 and is a mutational hotspot associated with diseases of electrical excitation in the heart and brain of humans (*Calhoun and Isom, 2014*; *Thomas et al., 2007*).

We next assessed the ability of βadp1 to abrogate β1 adhesion using the 1610 β1OX model. βadp1 significantly, persistently and dose-dependently decreased junctional resistance in confluent monolayers of 1610 β1OX cells, but not in 1610 parental cells (*Figure 3D*), which express negligible levels of β1. These results suggest that βadp1 selectively inhibited the intercellular adhesion function of β1 in this in vitro model. Similarly, βadp1 efficiently and dose-dependently inhibited the formation of intercellular interactions between sub-confluent 1610 β1OX cells forming into monolayers (*Figure 3E*), doing so at lower concentrations of peptide than necessary for established confluent

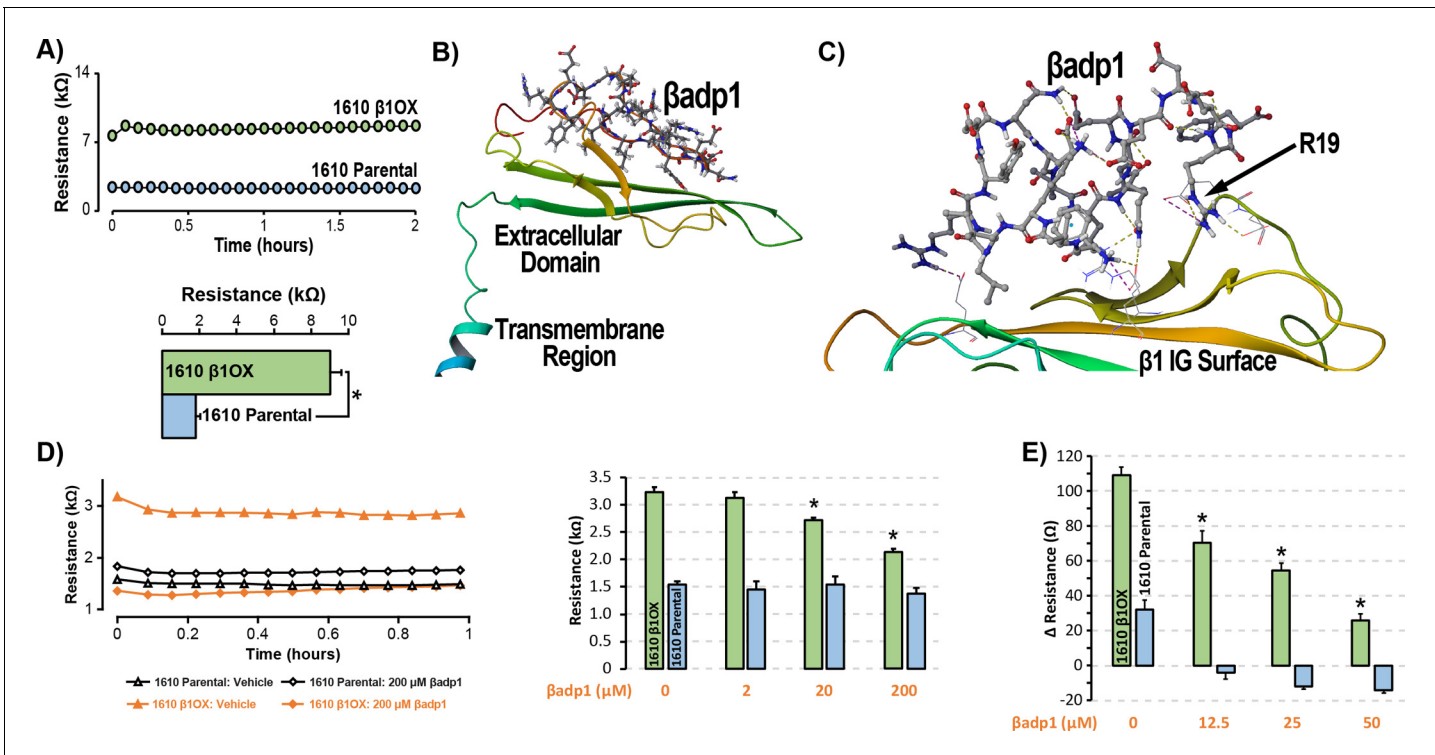

**Figure 3.** βadp1 – a novel inhibitor of β1-mediated adhesion. (**A**) Representative traces (top) and summary plot (bottom) of intercellular junctional resistance measured by ECIS in 1610 β1OX, and 1610 Parental cells (five experimental replicates with two technical replicates per experiment; $4 \times 10^4$ cells/well). (**B**) Homology model of the β1 ectodomain based on the β3 crystal structure, with the βadp1 sequence highlighted. (**C**) Docking of βadp1 with the β1 homology model in silico in a low-energy conformation with the adhesion surface of the β1 Ig loop. (**D**) Representative traces (left) and summary plot (right) demonstrating the effects of βadp1 on intercellular junctional resistance in 1610 β1OX and 1610 parental cells (five experimental replicates with two technical replicates per experiment, $10^4$ cells/well, *p<0.05 by 2-factor ANOVA). (**E**) Effects of βadp1 on the formation of intercellular interactions between sub-confluent 1610 β1OX and 1610 Parental cells, quantified as the change in resistance over 24 hr following plating in the absence/presence of βadp1.

The online version of this article includes the following figure supplement(s) for figure 3:

**Figure supplement 1.** R85D mutation abrogates βadp1 effects on β1-mediated adhesion.

**Figure supplement 2.** βadp1 effects on cell viability.

monolayers of β1 over-expressing cells. Neither the treatment of 1610 β1OX cells with βadp1-R85D (*Figure 3—figure supplement 1*), nor the treatment 1610 parental cells with βadp1 (*Figure 3D,E*) affected the establishment of cell-cell contacts, as assessed by ECIS. βadp1 showed no evidence of toxic effects on 1610 β1OX cells, or the mouse ventricular myocyte cell line H9C2 even at concentrations twice that used elsewhere in this study (*Figure 3—figure supplement 2*), suggesting that the peptide's action was unlikely to result from cellular toxicity.

## Inhibition of β1 adhesion results in de-adhesion of β1-enriched perinexal nanodomains

LSCM and STORM results described above demonstrate that perinexal ID regions show high enrichment for β1. Perfusion of βadp1 into Langendorff-perfused GP hearts had profound yet highly selective effects on the ultrastructure of these regions (*Figure 4*). A confocal image of a section from a GP ventricle perfused with biotinylated βadp1 (βadp1-b; 100 μM) in *Figure 4A* shows punctate βadp1-b signal (red) adjacent to Cx43 immunosignal (green), consistent with the peptide localizing in perinexal regions of the ID. Representative images (*Figure 4B*) obtained by transmission electron microscopy (TEM) illustrate close apposition between perinexal membranes in control tyrode-perfused GP ventricles and markedly wider spacing between perinexal membranes in βadp1-treated hearts, reflecting that βadp1 treatment induced a de-adhesion of the β1-enriched membranes of the perinexus.

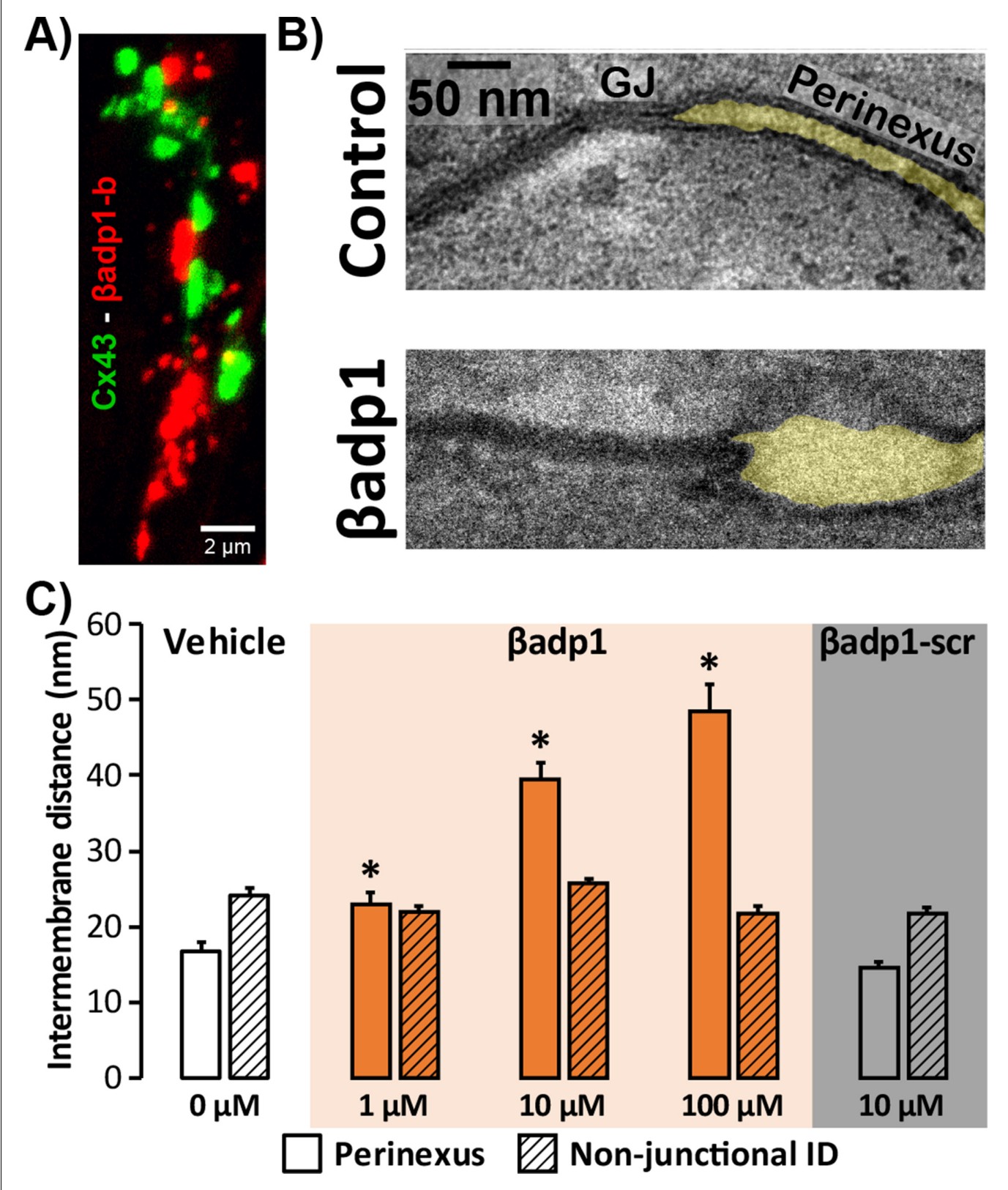

**Figure 4.** Modulating β1-mediated adhesion. (**A**) A representative confocal image from a GP LV perfused with biotinylated βadp1 (βadp1-b) shows Cx43 (green) and βadp1-b (red) immunosignals. (**B**) Representative TEM images of GJ and perinexi from hearts perfused with control Tyrode's solution

*Figure 4 continued on next page*

*Figure 4 continued*

(top) and βadp1 (100 µM; bottom). (C) A summary plot of inter-membrane distance at perinexal (open bars) and non-junctional ID (hatched bars) from hearts perfused with vehicle (white), βadp1 (orange), or a scrambled version of βadp1 (βadp1-scr; gray) [n = 3 hearts/dose, *p<0.05 by 2-factor ANOVA].

The online version of this article includes the following figure supplement(s) for figure 4:

**Figure supplement 1.** Perinexal ultrastructure in β1-null mice.

Overall, quantification of TEM images indicated that βadp1 increased inter-membrane distance in dose-dependent fashion within perinexi, but importantly, not at other ID locations distal from GJs (*Figure 4C*). In contrast, a scrambled control peptide (βadp1-scr; 10 µM) did not significantly affect inter-membrane spacing at perinexal or non-perinexal ID sites. These results indicate that βadp1 selectively, and dose-dependently, widened β1-enriched perinexi – a dehiscence of these nanodomains that would be consistent with the specific antagonism of β1-mediated adhesion within the ID by the inhibitory peptide. To further investigate the effects of loss of β1 function on perinexal ultrastructure, we examined ventricles from *Scn1b -/-* (β1-null) mice. β1-null ventricles displayed striking evidence of perinexal de-adhesion, with > 4 fold greater perinexal inter-membrane distances compared to wild-type (WT) littermates, while inter-membrane distance at non-perinexal ID sites was not different between the two genotypes (*Figure 4—figure supplement 1*). These results are consistent with the effects of βadp1 in GP ventricles.

## Inhibition of β1 adhesion selectively reduces GJ-associated sodium currents

In previous studies, we showed that acute interstitial edema increased perinexal membrane spacing at these $Na_V1.5$-enriched nanodomains and precipitated anisotropic conduction slowing (*Veeraraghavan et al., 2015*) – an effect on cardiac conduction that could only be explained by a computer model incorporating both ephaptic and electrotonic (i.e. GJ-based) coupling, and not by one incorporating electrotonic coupling alone. The level of perinexal de-adhesion caused by βadp1 was more marked than that prompted by experimentally induced edema. Thus, the aforementioned results suggested the potential of βadp1 as a tool for selectively probing the contribution of β1, and by extension, the hypothesized ephaptic mechanism, to action potential (AP) propagation between myocytes. As a first step to assess the effects of βadp1 on intercellular conduction of electrical impulse, we undertook voltage and current clamp studies on single myocytes isolated from GP hearts to determine whether the peptide affected AP and/or sodium channel function properties independent of whether cells where attached to one another. Neither βadp1, nor a scrambled control peptide (βadp1-scr), measurably altered the sodium current ($I_{Na}$; *Figure 5A–D*) or the duration or morphology of action potentials (APs; *Figure 5E–F*), suggesting that neither peptide affected the function of the $Na^+$ channel complex, or other ion channels, in isolated, non-contacting cells.

Next, we investigated whether selective targeting of the β1 adhesion domain affected $Na^+$ channel activity in cells forming adherent contacts with each other. The rationale for our approach was as follows: The ephaptic coupling hypothesis implies that $Na_V1.5$ channels on one cell's membrane can *trans*-activate $Na_V1.5$ channels on another cell membrane provided the two cells are separated by a sufficiently narrow extracellular cleft (*Mori et al., 2008*; *Lin and Keener, 2014*). Indeed, β1-mediated adhesion between cells may be critical to such a phenomenon by generating the close membrane apposition required for ephapse formation. Thus, modulating β1-mediated adhesion may alter the current from ID-localized $Na_V1.5$ in the presence of cell-to-cell contacts.

To test this hypothesis, we quantified $I_{Na}$ from cell-to-cell contact sites using scanning ion conductance microscopy (SICM) –guided smart patch clamp (SPC) in Cx43-EGFP-expressing neonatal rat ventricular myocyte (NRVM) cultures (*Figure 6*). Confocal immunolabeling of NRVMs revealed close association of β1 signals with Cx43 GJs at cell-to-cell contact sites (*Figure 6A*, *Figure 6—figure supplement 1*), consistent with observations from GP ventricles (*Figure 1*, *Figure 1—figure supplement 3*). Under control conditions, local $I_{Na}$ density at junctional sites proximal to Cx43-EGFP fluorescence, quantified by SPC, was significantly greater than that measured at non-junctional sites (*Figure 6—figure supplement 2B*), in line with previous reports of higher $I_{Na}$ density at the ID relative to the lateral membrane (*Lin et al., 2011*). Treatment with βadp1, but not βadp1-scr, significantly reduced $I_{Na}$ density at junctional sites (*Figure 6B–E*, *Figure 6—figure supplement 2C,D*).

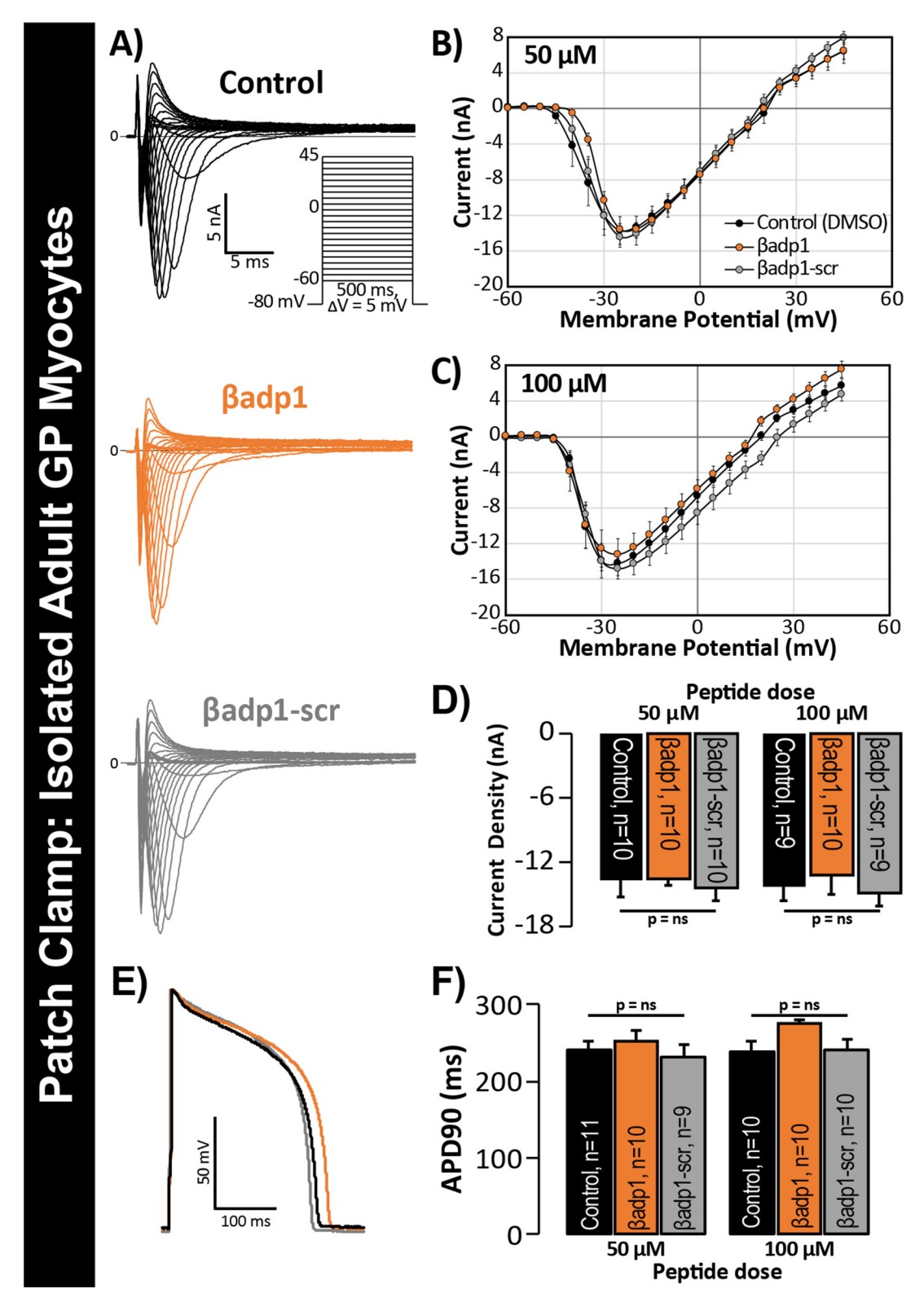

**Figure 5.** βadp1 effects on I_Na, APs. (**A**) Representative I_Na traces from myocytes during vehicle control, or treatment with βadp1 or βadp1-scr. Current-voltage relationships during treatment with (**B**) 50, or (**C**) 100 µM peptides. (**D**) Summary plots of I_Na density. (**E**) Representative AP traces and (**F**) AP duration during peptide treatment. Number of cells measured indicated on bar graphs.

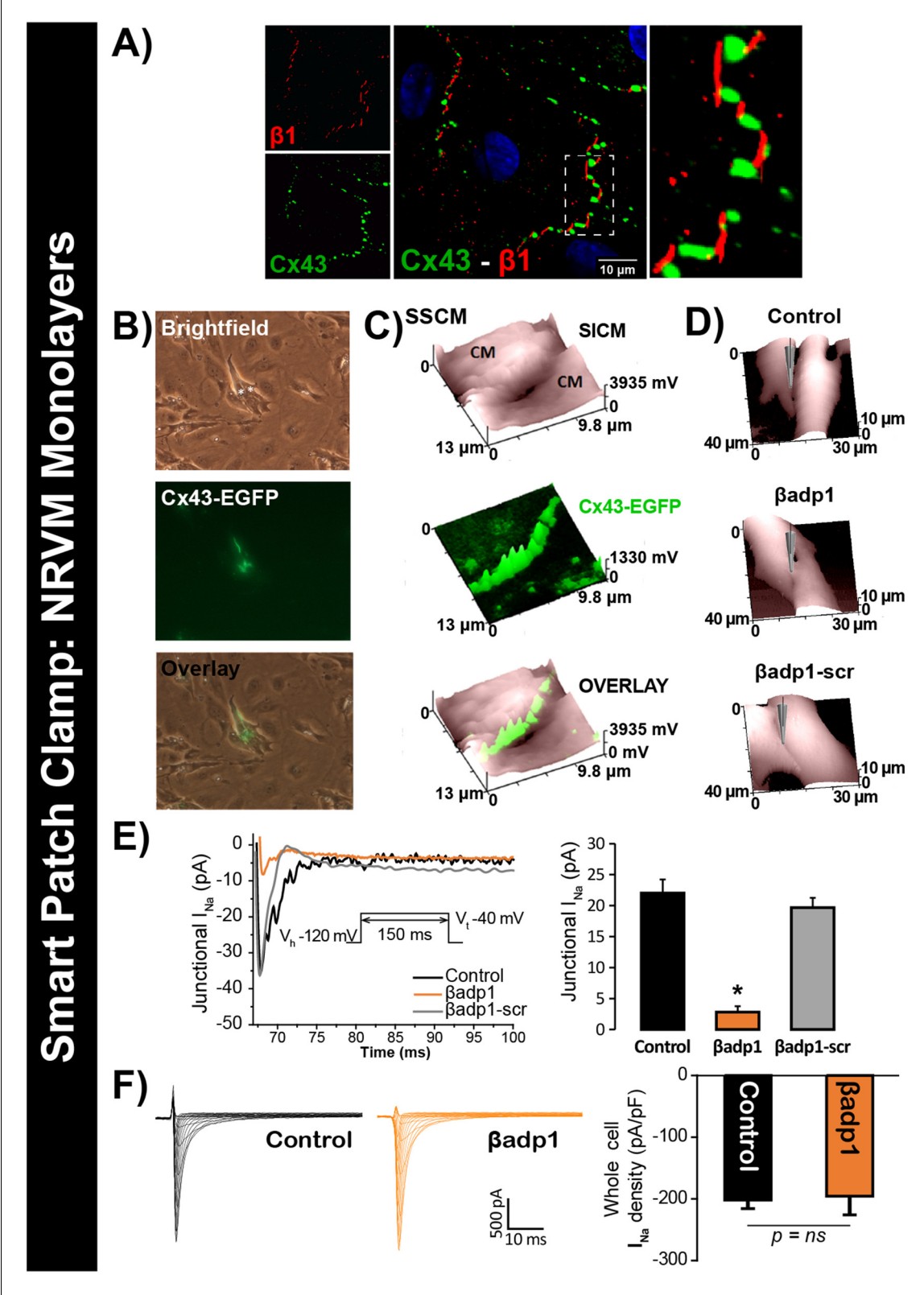

**Figure 6.** βadp1 effects on $I_{Na}$. (**A**) Representative confocal image of NRVMs labeled for Cx43 (red) and β1 (green). (**B**) Paired brightfield and fluorescence images along with an overlay demonstrate Cx43-EGFP fluorescence at cell-to-cell contacts. (**C**) Paired images of cell surface (SICM) and Cx43-EGFP fluorescence (surface scanning confocal microscopy; SSCM) at a cell-to-cell contact site. (**D**) Representative SICM images of cell-to-cell contacts under control conditions and following 30 min of treatment with βadp1 or βadp1-scr (50 μM) illustrate sites where $I_{Na}$ was measured using SPC.
*Figure 6 continued on next page*

*Figure 6 continued*

(E) Representative traces and summary plot of $I_{Na}$ from Cx43-EGFP fluorescence-adjacent junctional sites (Control: n = 12, βadp1: n = 8, βadp1-scr: n = 6; *p<0.05 vs control). (F) Representative traces and summary plot of whole-cell $I_{Na}$ density (n = 6/ group).

The online version of this article includes the following figure supplement(s) for figure 6:

**Figure supplement 1.** β1 expression in NRVMs.

**Figure supplement 2.** βadp1 effects on $I_{Na}$ at cell-to-cell contacts.

Currents at membrane sites distal from Cx43-GFP loci were not affected (*Figure 6—figure supplement 2E*). Importantly, and as was observed in isolated adult myocytes (*Figure 5D*), neither peptide altered whole cell $I_{Na}$ density (*Figure 6F*). These observations indicated that whilst disruption of β1-mediated adhesion caused localized changes in Cx43-GJ-adjacent $I_{Na}$ current density, global $I_{Na}$ density across the entire cell remained unaffected – results consistent with β1 maintaining the high density of *trans*-activating sodium channels at the perinexus necessary for the hypothesized ephaptic conduction mechanism.

## Inhibition of β1 adhesion induces proarrhythmic conduction slowing

To probe whether loss of β1 adhesion, and the consequent perinexal dehiscence and reduction in GJ-associated $I_{Na}$, affected cardiac conduction, we assessed the electrophysiologic impact of βadp1 treatment in Langendorff-perfused GP hearts. Representative electrocardiogram (ECG) traces in *Figure 7A* of intrinsic activity demonstrate prolongation of the QRS complex and the QT interval in the presence of βadp1 (orange trace) compared to control (black trace). Overall, βadp1, but not βadp1-scr, significantly prolonged both the QRS complex (*Figure 7B*) and the QT interval (*Figure 7C*), indicating that βadp1 impaired conduction. The QRS prolongation elicited by βadp1 occurred in a dose-dependent manner (*Figure 7D*). Further, βadp1 prolonged both the QRS complex and the QT interval to similar extents, suggesting that the ST segment and therefore, AP duration, was not prolonged.

Representative optical APs in *Figure 7E* demonstrate similar AP duration during control (black) and βadp1 (orange), consistent with ECG findings. Representative optical isochrone maps in *Figure 7F* show elliptical spread of activation from sites of unipolar epicardial pacing (white symbols). The maps also reveal crowding of isochrones, particularly along the transverse axis of propagation, in the presence of βadp1 (bottom) relative to control (top). This indicates preferential slowing of transverse conduction in the presence of βadp1. Summary plots in *Figure 7G* demonstrate that even low concentrations (10 μM) of βadp1 significantly slowed transverse conduction, thereby, increasing anisotropy, whereas βadp1-scr did not measurably affect conduction. In accordance with QRS prolongation, βadp1 reduced CV in a dose-dependent manner (*Figure 7H*): All doses tested (1–100 μM) preferentially decreased transverse conduction and increased anisotropy, while the higher concentrations (50 and 100 μM) also slowed longitudinal conduction. Importantly, the anisotropic conduction slowing induced by βadp1 proved proarrhythmic in a dose-dependent manner (*Figure 7A*, bottom trace; *Figure 7—figure supplement 1*): spontaneous ventricular tachycardias (VTs) were observed in the presence of βadp1 at doses (in μM) of 10 (1/3 hearts, p=0.2 vs. vehicle), 50 (2/3 hearts, p<0.05 vs. vehicle) and 100 (3/4 hearts, p<0.05 vs. vehicle). In contrast, no arrhythmias were observed in the presence of vehicle (DMSO; 0/3 hearts), or βadp1-scr (0/3 hearts).

In order to confirm that the impact of βadp1 reflected effects on AP conduction between cardiomyocytes, and was not due to effects on non-myocyte cells and tissues, we performed additional experiments assessing excitation spread in monolayers of human-induced pluripotent stem-cell-derived cardiomyocytes (iPSC-CMs). Immuno-confocal microscopy revealed enrichment of both $Na_V1.5$ and β1 adjacent to Cx43 at cell-to-cell contacts (*Figure 7—figure supplement 2A,B*), consistent with results in adult GP ventricles (*Figures 1* and *2*) and NRVMs (*Figure 6A*). In optical mapping experiments, treatment with βadp1, but not βadp1-scr, increased activation delay between equally spaced sites (*Figure 7—figure supplement 2C*) and slowed conduction in a dose-dependent manner (*Figure 7—figure supplement 2D*). Taken together with the results from GP ventricles, these data suggest that selective inhibition of β1-mediated adhesion slows conduction between cardiac myocytes in a cell-specific and pro-arrhythmic manner.

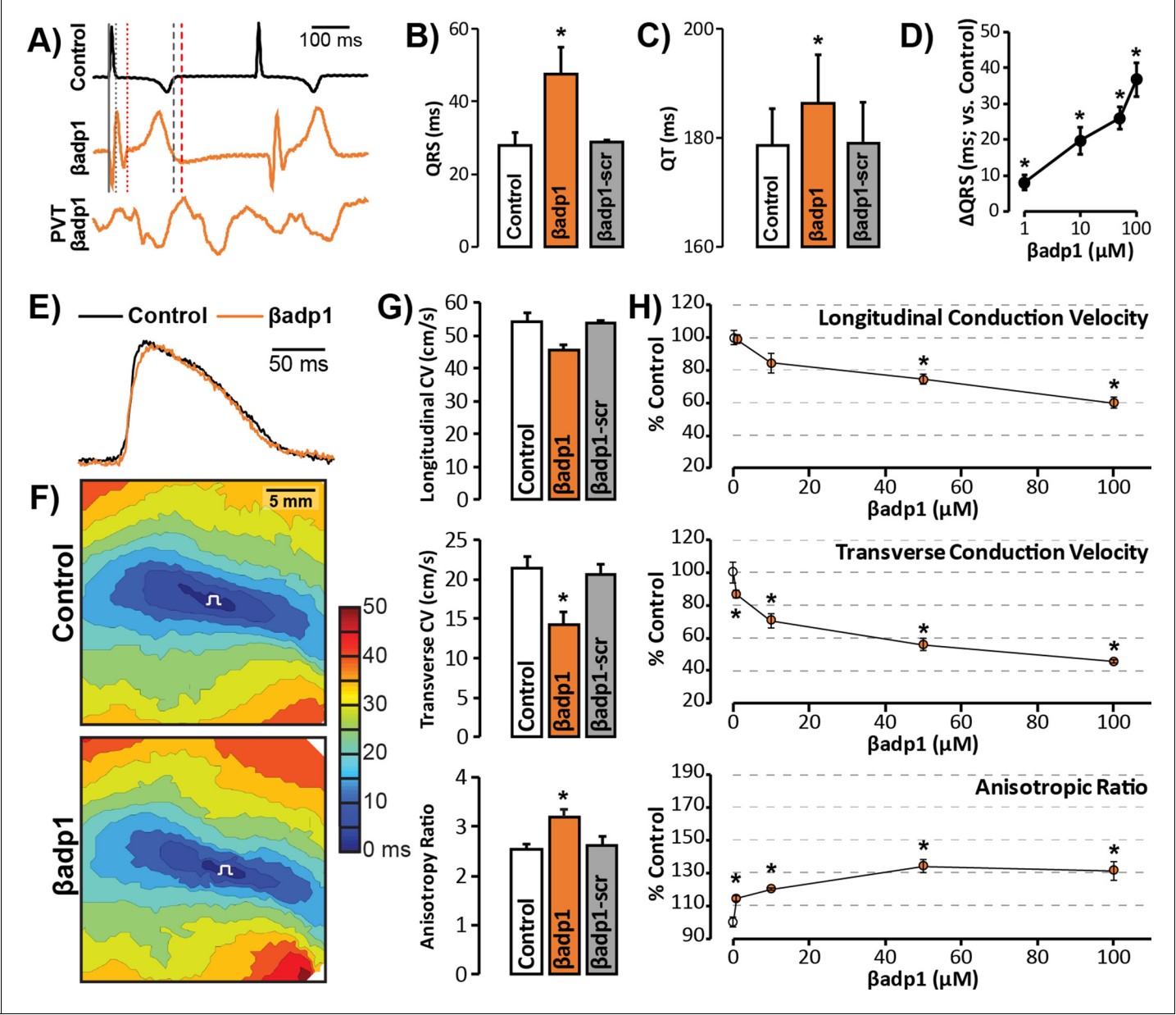

**Figure 7.** βadp1 effects on cardiac electrophysiology. (**A**) Representative traces of volume-conducted ECGs from GP ventricles during control (black) and βadp1 (10 μM; orange) perfusion. The solid gray vertical line marks the start of the QRS in both traces while gray and red dotted lines mark the end of the QRS in the control and βadp1 traces, respectively. Likewise, gray and red dashed lines mark the ends of the T wave in the control and βadp1 traces, respectively. The bottom orange trace shows an example of a polymorphic ventricular tachycardia (PVT) observed during βadp1 perfusion. Summary plots of (**B**) QRS duration and (**C**) QT interval during control, βadp1 (10 μM) and βadp1-scr (10 μM) perfusion (*p<0.05 vs. control). (**D**) Summary plot of QRS prolongation relative to control induced by different doses of βadp1 (*p<0.05 vs. control). Representative optical (**E**) APs and (**F**) isochrone maps of activation during unipolar epicardial pacing during control and βadp1 treatment (10 μM). The white symbols indicate the sites of pacing. (**G**) Summary plots of longitudinal conduction velocity ($CV_L$; top), transverse conduction velocity ($CV_T$; middle) and anisotropy ratio (AR; bottom) during control (white), βadp1 (10 μM; orange) and βadp1-scr (10 μM; gray) perfusion (*p<0.05 vs. control). (**H**) Summary plots of normalized $CV_L$ (top), $CV_T$ (middle) and AR (bottom) at different doses of βadp1 during pacing at 300 ms cycle length (n = 3 per dose per treatment, *p<0.05 by 2-factor ANOVA).

The online version of this article includes the following figure supplement(s) for figure 7:

**Figure supplement 1.** VT Incidence.

**Figure supplement 2.** βadp1 effects on conduction in iPSC-CMs.

## Discussion

In this study, we provide evidence that sodium channel complexes concentrated at the edge of GJs (i.e. in the perinexus) could provide a structural basis for ephaptic conduction in the heart (*Figure 8*). Importantly, we find that the extracellular adhesion domain of the sodium channel β1 subunit may be critical for the generation of the close (<30 nm) cell-cell interactions necessary for this mechanism to operate. Selective inhibition of β1-mediated adhesion profoundly disrupts the structure of GJ-adjacent perinexi, causing them to dehisce/de-adhere, and swell beyond limits that would theoretically support ephaptic conduction, with attendant loss of perinexal sodium currents, anisotropic conduction slowing and arrhythmias. The clinical translational significance of perinexal dehiscence is further emphasized by our recent report that it is associated with the occurrence of atrial fibrillation in human patients (*Raisch et al., 2018*). Interestingly, the physical extent of the β1 extracellular domain predicted by our molecular homology model (*Figure 3B*) is 7–15 nm, depending on relaxation state of the domain. Thus, the span of two β1 extracellular domains on apposed cell membranes in trans homophilic interaction concurs with the EM-based measurements of average perinexal width (15–30 nm) in untreated hearts that we report here, and elsewhere (*Veeraraghavan et al., 2015*).

Ephaptic coupling has been observed in neural (*Anastassiou et al., 2011*; *Bokil et al., 2001*; *Chan et al., 1988*; *Faber and Korn, 1989*; *Jefferys, 1995*; *Su et al., 2012*; *Voronin, 2000*), and other (*Klaassen et al., 2012*; *Young, 2007*) tissues, and has long been hypothesized to occur in the heart (*Mori et al., 2008*; *Hichri et al., 2018*; *Kucera et al., 2002*; *Lin and Keener, 2013*; *Sperelakis, 2002*). Early work on ephaptic coupling in the heart was motivated by the fact that GJs, which are abundant in mammalian hearts, are over two orders of magnitude less frequent in other chordates, especially birds (*Martínez-Palomo and Mendez, 1971*; *Shibata and Yamamoto, 1979*).

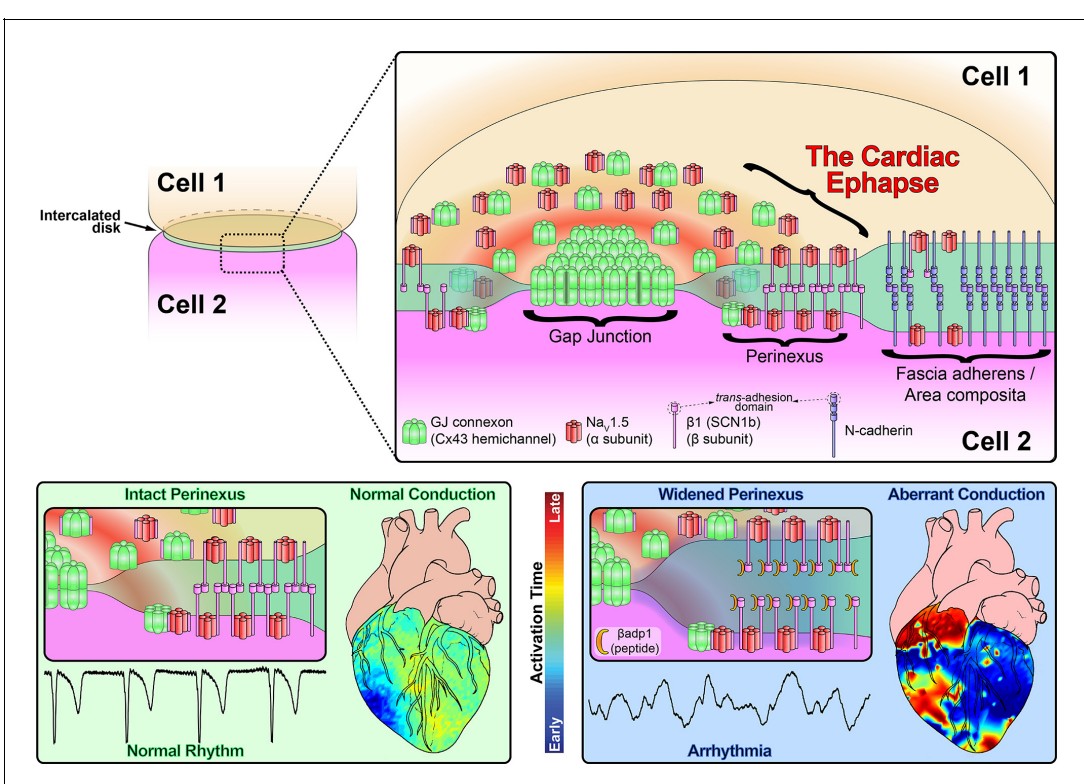

**Figure 8.** Schematic diagram of the cardiac ephapse. Top. A schematic diagram illustrating the organization of Na$_V$1.5, β1, Cx43 and N-cad to different ID nanodomains. Note that plicate (fascia adherens/area composita) and interplicate (GJ, perinexus) regions of the ID are displayed side-by-side for convenience of representation, although they are, in reality, oriented along perpendicular axes. Bottom left. Illustration of an intact perinexus with close membrane apposition displayed with an ECG trace of intrinsic activity and illustration of normal intrinsic activation sequence represented on a diagram of the heart. The earliest sites of activation are represented in blue, and the latest in red as illustrated by the color bar. Bottom right. Illustration of a widened perinexus resulting from βadp1 inhibition of β1-mediated adhesion with an ECG trace of a resulting PVT and illustration of arrhythmic activation sequence represented on a diagram of the heart.

Mathematical modeling suggests that this level of GJ coupling is unlikely to reliably support the beat-to-beat conduction of AP long-term, thus indicating that an alternate paradigm for AP conduction may operate in bird hearts. Nevertheless, the investigation of ephaptic conduction has remained largely theoretical and the hypothesis has drawn considerable skepticism over the years. The present study identifies a potential structural unit within the ID located at the edge of GJs – the ephapse – that may provide a mechanistic basis for ongoing studies of this long-theorized mechanism of cardiac AP conduction.

Subcellular distribution (and thus, nano-structural context) has emerged as an important modulator of Na$_V$1.5 function (*Hichri et al., 2018*; *Petitprez et al., 2011*; *Lin et al., 2011*; *Chen-Izu et al., 2015*; *Eichel et al., 2016*; *Hund and Mohler, 2014*; *Makara et al., 2014*): Na$_V$1.5 has been reported to be divided into lateral and ID-localized pools based on its sarcolemmal localization, differences in scaffolding partners, and current properties. Interestingly, Hichri and colleagues recently demonstrated that Na$_V$1.5 behavior is influenced by channel clustering, and localization facing narrow extracellular clefts (*Hichri et al., 2018*), raising the possibility of Na$_V$1.5 pools playing different, ultrastructure dependent roles. Here, using super-resolution STORM-RLA (*Veeraraghavan and Gourdie, 2016*), we further categorize ID-localized Na$_V$1.5 into a 'perinexal pool' adjacent to Cx43 and a 'plicate pool' that co-distributes with N-cad. The perinexal pool is in agreement with our demonstration of Na$_V$1.5 enrichment proximal to Cx43 GJs (*Veeraraghavan et al., 2015*; *Veeraraghavan and Gourdie, 2016*; *Rhett et al., 2012*; *Veeraraghavan et al., 2016*), while the plicate pool is consistent with the results of Leo-Macias and colleagues who reported Na$_V$1.5 co-distributing with N-cadherin-positive adherens junctions within the ID (*Leo-Macias et al., 2016*). Whilst adjacent cell membranes are very closely apposed (5–15 nm) within the perinexus (*Veeraraghavan et al., 2015*), they are further apart (63 nm) in the vicinity of adhesion complexes within plicate ID regions (*Leo-Macías et al., 2015*). Given that theoretical studies indicate that membrane spacing of ≤30 nm would be necessary to support ephaptic coupling (*Mori et al., 2008*; *Greer-Short et al., 2017*), the plicate Na$_V$1.5 pool is unlikely to play a role in ephaptic coupling. Nonetheless, whilst closely approximated, *trans*-activating sodium channels at GJ perinexi may act to 'trigger' a local depolarization of interplicate membrane, the function of the nearby pools of plicate Na$_V$1.5 channels could be to provide *cis*-activated excitatory current, contributing to the spread of depolarization through the downstream myocyte.

The approach we took to developing a competitive inhibitor of β1 was based on a methodology successfully applied by independent groups to two other Ig domain adhesion molecules that share homologies with β1: N-cad (*Williams et al., 2000a*), and desmoglein (*Schlipp et al., 2014*). This strategy involved generating a peptide mimetic of the extracellular adhesion sequence of the β1 Ig domain. In ECIS experiments, β1 overexpression conferred strong adhesion to 1610 cells, which was selectively and dose-dependently inhibited by βadp1 treatment. When perfused into GP ventricles, βadp1 prompted dose-dependent increases in inter-membrane distance, selectively within perinexi. Similar effects were not observed in hearts treated with a scrambled control peptide, βadp1-scr. The conspicuous de-adhesion of perinexal membranes prompted by βadp1 in GP hearts, and by germline loss of *Scn1b*/β1 in mice was selective to perinexal nanodomains, with inter-membrane spacing at other ID locations (e.g. within adherens junctions and desmosomes) remaining unaffected in both GPs and mice. These results, along with the demonstration that βadp1 treatment had no effects on whole-cell I$_{Na}$, but results in localized loss of I$_{Na}$ adjacent to Cx43-GFP GJs, attest to the selectivity of the novel inhibitor of β1-mediated adhesion. Further, these findings suggest that the increases in extracellular cleft width adjacent to Cx43 GJ induced by inhibition of β1-mediated adhesion likely compromises *trans*-activation of Na$^+$ channels within these nanodomains. Ongoing studies would usefully determine the basis of this localized down-regulation of channel activity. Crucially, the fact that βadp1 treatment of myocyte monolayers elicits these localized changes in I$_{Na}$ within GJ-adjacent nanodomains, without alteration of whole-cell I$_{Na}$, indicates the possibility of local remodeling of Na$^+$ channel complexes from perinexal nanodomains.

Examining the functional impact of βadp1 treatment on Langendorff-perfused GP ventricles, we observed prolongation of the QRS complex, pointing to conduction slowing. The QT interval was also prolonged by a similar degree as the QRS complex, indicating no change in ST segment duration. This is consistent with the absence of APD changes following βadp1 treatment in both isolated myocytes and in intact ventricles. The contrast between these results, and those reported in mice lacking β1 (β1-null), where QT prolongation was associated with prolonged AP duration (*Lopez-*

*Santiago et al., 2007*), likely stem from the differences between acutely and selectively inhibiting β1-mediated adhesion (as in the present study), versus chronic genetic ablation of β1 expression. As yet, conduction velocity measurements have not been reported from the ventricles of β1-nulls, which in light of the data reported herein would be a useful addition to our understanding of cardiac conduction mechanisms.

In keeping with the observed QRS prolongation, βadp1 treatment dose-dependently slowed transverse conduction in intact ventricles, increasing the anisotropy of conduction, and precipitating spontaneous arrhythmias. We have previously demonstrated in silico that anisotropic conduction slowing secondary to a reduction of $I_{Na}$ at the ID results from compromised ephaptic coupling (*Veeraraghavan et al., 2015*): Briefly, a wavefront traveling transverse to the fiber axis encounters more cell-cell junctions per unit distance than one traveling longitudinally. Therefore, compromised cell-cell coupling via perinexal nanodomains would have a greater impact on transverse conduction than on longitudinal conduction. Thus, the anisotropic nature of conduction slowing caused by βadp1 is consistent with disruption of ephaptic communication (*Plonsey and Barr, 2007*), and identifies a hitherto unanticipated role for β1-mediated adhesion in the heart. Taken together, these data lend further support to the hypothesis that β1 adhesion mediates close apposition between $Na_V1.5$-rich membranes within the perinexus, facilitating ephaptic coupling between cardiac myocytes (*Figure 8*). In this context, the fact that βadp1 mimics a sequence in β1 (i.e. its Ig domain adhesion sequence) that is a 'hot spot' for disease-causing mutations, including mutations associated with atrial arrhythmia (R85H) (*Watanabe et al., 2009*), conduction disease (R85H) (*Watanabe et al., 2009*), epilepsy (R85H, R85C) (*Scheffer et al., 2007*), Brugada syndrome (E87Q) (*Hasdemir et al., 2015*), and febrile epilepsy (I70_E74del) (*Audenaert et al., 2003*) lends further support to the importance of β1-mediated adhesion to electrical propagation in the heart, and beyond. Interestingly, a single amino acid substitution of R to D at the R85 'hot spot' within βadp1 significantly abrogated the cell adhesion-inhibitory effects of the peptide. These data suggest that disease-causing mutations associated with this locus may mediate their effects via perinexal disruption.

In summary, we provide experimental evidence suggesting that ephaptic conduction may play a role in the heart, identifying hitherto unanticipated assignments for both $Na_V1.5$ and β1 in this phenomenon. Although limitations of current technology prevent direct interrogation of electrophysiology at the nanoscale, which would provide direct proof of the ephaptic mechanism, our results strongly suggest a role for it in normal cardiac physiology. Thus, it has not escaped our notice that modulation of β1-mediated adhesion could represent a novel clinical target for the amelioration of cardiac arrhythmias. Increased spacing of perinexal membranes, whether resulting from inhibition of β1-mediated adhesion, or acute interstitial edema, is strongly associated with ventricular arrhythmias. Given that humans with atrial fibrillation also demonstrate perinexal dehiscence (*Raisch et al., 2018*), these data suggest that drugs that act to stabilize and/or enhance β1-mediated adhesion within IDs could have anti-arrhythmic potential.

## Materials and methods

### Animal preparations
Hearts were isolated from male retired breeder guinea pigs (GPs), and 19-day-old *Scn1b* -/- mice, perfused as Langendorff-preparations, or preserved for cryosectioning or transmission electron microscopy (TEM).

### Confocal and super-resolution microscopy
Immunofluorescent staining was performed, as previously described (*Veeraraghavan et al., 2015*; *Veeraraghavan and Gourdie, 2016*; *Veeraraghavan et al., 2016*), on 5 µm cryosections of tissue, and monolayers of cells fixed with paraformaldehyde (2%; 5 min at room temperature). Confocal imaging was performed using a TCS SP8 confocal microscope, while super-resolution STochastic Optical Reconstruction Microscopy (STORM) was performed using a Vutara 350 microscope. Single molecule localization data from STORM was quantitatively analyzed using STORM-RLA as previously described (*Veeraraghavan and Gourdie, 2016*).

## Electric cell-substrate impedance spectroscopy (ECIS)

Intercellular junctional resistance was quantified using an ECIS Zθ system in 1610 cells stably overexpressing β1 (1610 β1OX) as well as in parental 1610 cells (1610 Parental).

## Transmission electron microscopy (TEM)

TEM images of the ID, particularly GJs and mechanical junctions, were obtained at 100,000x magnification on a JEOL JEM-1400 electron microscope.

## Isolated myocyte electrophysiology

Myocytes were isolated from adult GP hearts by enzymatic dispersion. Action potentials and $Na^+$ currents were recorded by current clamp, and voltage clamp, respectively.

## Surface scanning confocal microscopy (SSCM) and smart patch clamp (SPC)

Neonatal rat ventricular myocytes (NRVMs) were isolated by a combination of mechanical dissociation, and enzymatic degradation, and cultured as monolayers. SSCM, which combines scanning ion conductance microscopy (SICM) with confocal microscopy, was used to concurrently image membrane topology and fluorescent signals. Local $I_{Na}$ from regions thus identified were recorded by SPC as previously described (*Bhargava et al., 2013*). Additionally, whole-cell $I_{Na}$ was measured from NRVMs by voltage clamp.

## Optical mapping and electrocardiography

Whole heart electrophysiology was assessed by volume-conducted electrocardiograms (ECG), and optical voltage mapping using the voltage-sensitive dye, di-4-ANEPPS (15 μM), as previously described (*Veeraraghavan et al., 2015*; *Veeraraghavan et al., 2016*; *Veeraraghavan and Poelzing, 2008*).

## Detailed methods

The investigation conforms to the *Guide for the Care and Use of Laboratory Animals* published by the US National Institutes of Health (NIH Publication No. 85–23, revised 1996). All animal study protocols (15-130, 15-134, 12-140) were approved by the Institutional Animal Care and Use Committee at the Virginia Polytechnic University.

### Animal preparations

Male retired breeder, albino Hartley guinea pigs (GPs; Hilltop Lab Animals, Scottsdale, PA, USA;~900–1200 g, 13–20 months old) were anesthetized with 5% isoflurane mixed with 100% oxygen (3 l/min). After loss of consciousness, anesthesia was maintained with 3–5% isoflurane mixed with 100% oxygen (4 l/min). Once the animal was in a surgical plane of anesthesia, the heart was excised and the ventricles were either fixed for transmission electron microscopy (TEM), frozen for cryosectioning/Western immunoblotting or perfused (at 40–55 mm Hg) as Langendorff preparations with oxygenated Tyrode's solution (containing, in mM: $CaCl_2$ 1.25, NaCl 140, KCl 4.5, dextrose 5.5, $MgCl_2$ 0.7, HEPES 10; pH adjusted to 7.41 with ~5.5 mM NaOH) at 37˚C as previously described (*Poelzing and Veeraraghavan, 2007*; *Veeraraghavan et al., 2015*; *Veeraraghavan et al., 2016*; *Veeraraghavan and Poelzing, 2008*). Additionally, *Scn1b* -/- (β1-null) and *Scn1b* +/+ (wild-type; WT) mice were generated as previously described (*Lopez-Santiago et al., 2007*), and their ventricles were processed for TEM as noted above. Murine tissue samples were collected from 19 day-old mice, because the loss of β1 is lethal from the 2nd week of life onwards.

### Fluorescent Immunolabeling

Immunofluorescent staining was performed, as previously described (*Veeraraghavan et al., 2015*; *Veeraraghavan et al., 2016*; *Veeraraghavan and Gourdie, 2016*), on 5 μm cryosections of tissue, and monolayers of cells fixed with paraformaldehyde (2%; 5 min at room temperature). Samples were labeled with our novel rabbit polyclonal antibodies against either $Na_V1.5$ (Epitope: $_{1996}$HSED LADFPPSPDRDRESIV$_{2016}$) or β1 (Epitope: $_{44}$KRRSETTAETFTEWTFR$_{60}$). Validation results for these antibodies are presented in *Figure 1—figure supplement 1*. In some cases, samples were co-

labeled with a mouse monoclonal antibody against either connexin43 (Cx43; Millipore MAB3067, 1:250) or N-cadherin (N-cad; BD Biosciences 610920, 1:100). For confocal microscopy, samples were then labeled with goat anti-rabbit AlexaFluor 568 (1:4000; ThermoFisher Scientific, Grand Island, NY) and goat anti-mouse AlexaFluor 633 (1:4000; ThermoFisher Scientific, Grand Island, NY) secondary antibodies. For super-resolution STochastic Optical Reconstruction Microscopy (STORM), samples were labeled with goat anti-rabbit Alexa 647 (1:4000) and donkey anti-mouse Cy3b (1:100) secondary antibodies (ThermoFisher Scientific, Grand Island, NY) and stored in Scale U2 buffer (*Hama et al., 2011*) for 48 hr at 4°C.

## Confocal microscopy

Confocal imaging was performed using a TCS SP8 laser scanning confocal microscope equipped with a Plan Apochromat 63x/1.4 numerical aperture oil immersion objective and a Leica HyD hybrid detector (Leica, Buffalo Grove, IL). Individual fluorophores were imaged sequentially with the excitation wavelength switching at the end of each frame.

## Western Immunoblotting

Whole cell lysates were prepared from frozen GP ventricles and 1610 cells; membrane lysates were prepared from brains of β1 -/- (β1-null) (*Lopez-Santiago et al., 2007*), and WT littermate mice as previously described (*Veeraraghavan and Poelzing, 2008*; *Rhett et al., 2011*). These were electrophoresed on 4–15% TGX Stain-free (for Na$_V$1.5) or 12% Bis-Tris (for β1) gels (BioRad, Hercules, CA) before being transferred onto a polyvinylidene difluoride (PVDF) membrane. The membranes were probed with our novel rabbit polyclonal antibodies against Na$_V$1.5 or β1, followed by a goat anti-rabbit HRP-conjugated secondary antibody (JacksonImmuno, West Grove PA). Signals were detected by chemiluminescence using SuperSignal West Dura Extended Duration Substrate (ThermoFisher Scientific, Grand Island, NY) and imaged using a Chemidoc MP imager (BioRad, Hercules, CA). For the GP ventricle and 1610 cell lysates, prep gels with a single 700 μl well were used, and following transfer, vertical strips cut from the PVDF membrane to be probed with different antibody solutions. Standard 12-well gels were used for the mouse brain lysates so that β1-null and WT mouse lysates could be run in alternating lanes.

## Dot blot

Stock antigen peptides were diluted (Na$_V$1.5 in TBST, β1 in DMSO) to 0.5 μg /μl followed by serial dilutions to 1:10 and 1:100. PVDF membranes were rehydrated in methanol, water, and then TBST before being transferred to blot paper to absorb excess moisture. Peptide or control solutions (2 μl) were spotted onto the damp membranes. The membranes were air-dried, rehydrated, and blocked for 1 hr at room temperature (RT). Primary antibodies were added and incubated overnight at 4°C. The membranes were washed in TBST (1 × 10 s, 2 × 5 min) followed by incubation with goat anti-rabbit horseradish peroxidase at RT for 1 hr. Antibody-peptide interactions were detected using chemiluminescence substrate Super Signal West Femto (ThermoFisher Scientific, Grand Island, NY) per manufacturer's instructions. The chemiluminescence signal was digitally captured using a Chemi-Doc MP imager (BioRad, Hercules, CA).

## STORM-based relative localization analysis (STORM-RLA)

STORM imaging was performed using a Vutara 350 microscope equipped with biplane 3D detection (*Deschout et al., 2014*; *Juette et al., 2008*; *Mlodzianoski et al., 2009*), and fast sCMOS imaging achieving 20 nm lateral and 50 nm axial resolution. Volumes imaged spanned between 10 × 10 μm and 15 × 15 μm along the x-y plane and 3–5 μm in the z-dimension. Localization of particles was accomplished with a precision of 10 nm. Registration of the two-color channels was achieved using a transform calculated from the localized positions of several TetraSpeck Fluorescent Microspheres (ThermoFisher Scientific, Carlsbad, CA) scattered throughout the field of view, similar to a previously described approach (*Churchman and Spudich, 2013*). The images were quantitatively analyzed using STORM-RLA as previously described (*Veeraraghavan and Gourdie, 2016*).

## Heterologous β1 expression in 1610 cells

In order to probe β1-mediated adhesion, Chinese hamster lung 1610 cells (ATCC CRL-1657; tested negative for mycoplasma), which do not endogenously express β1 (*Isom et al., 1995*), were used as a heterologous expression system. Studies were conducted in 1610 cells stably overexpressing β1 (1610 β1OX) as well as in parental 1610 cells (1610 Parental). The species identity of the 1610 cells was confirmed by COI assay, and the cell-type identity confirmed based on morphological characteristics.

## Cell viability assay

The effects of peptide treatment on the viability of cells was assessed using a colorimetric assay (Cell Proliferation Reagent WST-1, Sigma-Aldrich, St. Louis, MO). Briefly, H9C2 cells (ATCC CRL-1446; tested negative for mycoplasma) were cultured at 37°C and 5% $CO_2$ in microplates (96-well, flat bottom) with each well seeded with $5 \times 10^4$ cells in 100 µl culture media containing peptide or vehicle. After 24 hr, WST-1 (10 µl/well) was added, and allowed to incubate for 4 hr at 37°C and 5% $CO_2$. The microplate was thoroughly shaken for one minute before max absorption of 440 nm was quantified using a plate reader (SpectraMax i3) with background subtracted by reference wavelength of 600 nm. Cell viability was calculated as the ratio of corrected signal from peptide-treated samples to corrected signal from vehicle-treated controls (reported as a percentage). The species identity of the H9C2 cells was confirmed by COI assay, and the cell-type identity confirmed based on morphological characteristics.

## Electric cell-substrate impedance spectroscopy (ECIS)

The resistance of monolayers of 1610 cells was quantified using an ECIS Zθ system (Applied Biophysics, Troy, NY) over a range of 62.5 to 4000 Hz, previously demonstrated to reflect intercellular junctional resistance (*Moy et al., 2000*; *Tiruppathi et al., 1992*). Briefly, 1610 cells (400 µl/well, $2$–$10 \times 10^4$ cells/ml) were plated on 8W10E + 8-well dishes (Applied Biophysics, Troy, NY) where each well was equipped with 40 gold electrodes, and their impedance measured every 90 s for 3 hr. Resistance was quantified as the real component of the impedance and intercellular junctional resistance calculated as the average resistance between 62.5 and 4000 Hz.

## Transmission electron microscopy (TEM)

Cubes of tissue (1 mm side) were taken from the anterior LV free wall of GP and mouse hearts and fixed overnight in 2% glutaraldehyde at 4°C. TEM images of the ID, particularly GJs and mechanical junctions, were obtained at 100,000x magnification on a JEOL JEM-1400 electron microscope. Intermembrane distance at perinexal and non-perinexal sites was quantified using ImageJ (NIH, http://rsbweb.nih.gov/ij/) as previously described (*Veeraraghavan et al., 2015*).

## Adult myocyte isolation

Myocytes were isolated from LV-free wall of GP hearts using the enzymatic dispersion technique described previously (*Wan et al., 2005*). They were re-suspended in 10 ml of Dulbecco modified Eagle medium, stored at RT and used within 24 hr of isolation. Treatments were applied to myocytes 60 min prior to measurements.

## Action potentials (APs)

Patch clamping was performed in current clamp mode to record APs (*Nassal et al., 2016*). Briefly, cells were superfused in a chamber continuously perfused with Tyrode's solution composed of (in mM) NaCl 137, KCl 5.4, $CaCl_2$ 2.0, $MgSO_4$ 1.0, Glucose 10, and HEPES 10, with pH adjusted to 7.35 with NaOH. Patch pipettes were pulled from borosilicate capillary glass and lightly fire-polished to 0.9–1.5 MΩ resistance when filled with electrode solution composed of (in mM) aspartic acid 120, KCl 20, NaCl 10, $MgCl_2$ 2, HEPES 5, pH 7.3. Myocytes were paced in current clamp mode with a 5 ms pulse at 1.5–2 times the diastolic threshold, at a cycle length of 1000 ms. The experiments were performed at 35°C. Command and data acquisition were operated with an Axopatch 200B patch clamp amplifier controlled by a personal computer using a Digidata 1200 acquisition board driven by pCLAMP 7.0 software (Axon Instruments, Foster City, CA).

## Whole-cell sodium current ($I_{Na}$) recordings in adult myocytes

$Na^+$ currents were recorded by ruptured-patch whole cell voltage clamp at RT (*Hoshi et al., 2014*). Microelectrodes were filled with a solution of (in mM) CsF 120, MgCl$_2$ 2, HEPES 10, EGTA 11 and brought to a pH of 7.3. Isolated myocytes were placed in the solution containing (in mM) NaCl 25, N-methyl D-glucamine 120, CsCl 5, MgCl$_2$ 1, NiCl$_2$ 1, glucose 10, HEPES 10, pH 7.3. $I_{Na}$ was elicited from a holding potential of −80 mV with depolarizing voltage pulses from −60 mV to 45 mV for 500 ms. Current density (in pA/pF) was calculated from the ratio of current amplitude to cell capacitance.

## Neonatal rat ventricular myocyte (NRVM) isolation

NRVM isolation procedures conformed to the UK Animal Scientific Procedures Act 1986. Ventricles were isolated from one-day-old rat pups anesthetized with a lethal dose of isoflurane, sectioned into small cubes, and processed using a combination of mechanical dissociation (gentleMACS) and enzymatic degradation (neonatal heart dissection kit; Miltenyi Biotec, Bergisch Gladbach, Germany). The resulting cell suspension was filtered and ventricular myocytes were plated onto glass-bottom dishes (MatTek Corp., Ashland, MA) in M199 supplemented with newborn calf serum (10%), vitamin B12, glutamate and penicillin/streptomycin (1%). Myocytes were allowed to grow and establish connections for 3–4 days in vitro. For viral transfection, cells were treated for 2–3 days in vitro with pLP-Adeno-X-CMV Cx43-EGFPN1 construct (advenovirus type V E1). Peptide treatments were applied to cell monolayers 60 min prior to $I_{Na}$ measurements.

## Surface scanning confocal microscopy (SSCM)

SSCM combines scanning ion conductance microscopy (SICM) with confocal microscopy, with the laser being directed to the tip of the micropipette (*Gorelik et al., 2002*), to concurrently obtain a fluorescence image of the cell surface while capturing the topography of the exact same area. Briefly, a pair of NRVMs transduced with Cx43-EGFP was moved along the z axis while scanning along the x and y axes, using a three-axis piezo-translation stage. The cells' surface was maintained at a constant distance from the nanopipette to enable non-contact capture of cell surface topography. A laser was passed up a high numerical aperture objective so that it was focused at the tip of the nanopipette, and a pinhole was positioned at the image plane, thus, placing the confocal volume just below the pipette.

## SICM-guided smart patch clamp (SPC)

A variant of SICM, called hopping probe ion conductance microscopy, was combined with cell-attached recordings of cardiac sodium channels, as previously described (*Bhargava et al., 2013*), to assess $I_{Na}$ at cell-to-cell contact sites from NRVM monolayers. Currents were recorded in cell-attached mode using an Axopatch 200A/B patch-clamp amplifier (Molecular Devices, Sunnyvale, CA), and digitized using a Digidata 1200B data acquisition system and pClamp 10 software (Axon Instruments; Molecular Devices, Sunnyvale, CA). Briefly, cell-to-cell junctional sites were identified using a sharp scanning nano-probe (40–50 MΩ), followed by controlled increase of pipette diameter (20–25 MΩ) for capture of active sodium channel clusters. The external solution contained (in mM): KCl 145, Glucose 10, HEPES 10, EGTA 2, MgCl$_2$ 1, and, CaCl$_2$ 1 (300 mOsm and pH 7.4), while the internal (pipette) solution contained (in mM) NaCl 135, TEA-Cl 20, CsCl 10, 4AP 10, Glucose 5.5, KCl 5.4, HEPES 5, MgCl$_2$ 1, CaCl$_2$ 1, NaH$_2$PO$_4$ 0.4 and CdCl$_2$ 0.2 (pH 7.4). The voltage-clamp protocol consisted of sweeps to test potentials ranging from −70 to +30 mV from a holding potential of −120 mV.

## Whole-cell $I_{Na}$ recordings in NRVMs

Whole-cell $I_{Na}$ was recorded from NRVMs in low-sodium extracellular solution containing the following (in mM): CsCl 130, NaCl 11, Glucose 10, HEPES 10, MgCl$_2$ 2, CaCl$_2$ 0.5, CdCl$_2$ 0.3, adjusted to 7.4 with CsOH. The intracellular (pipette) solution contained (in mM): Cesium methanesulfonate (CsMeS) 100, CsCl 40, HEPES 10, EGTA 5, MgATP 5, MgCl$_2$ 0.75, adjusted to pH 7.3 with CsOH. Pipettes were pulled from the borosilicate glass microelectrodes and had a resistance of 3–4 MΩ. Sweeps were initiated from the holding potential of −100 mV to test potentials ranging from −75 to + 20 mV in 5 mV increments. Peak current was measured between −30 and −40 mV. Whole cell

capacitance ranged between 9 and 16 pF. Current density (in pA/pF) was calculated as the ratio of the peak current to cell capacitance.

## Electrocardiography

A volume-conducted electrocardiogram (ECG) was collected from Langendorff-perfused GP ventricles, as previously described (*Veeraraghavan et al., 2015*; *Veeraraghavan et al., 2016*; *Veeraraghavan and Poelzing, 2008*), using silver chloride electrodes placed in the bath and digitized at 1 kHz. The incidence of ventricular tachycardias (VTs; defined as three or more consecutive, non-paced heartbeats with a cycle length shorter than 130 ms) was quantified; all VTs observed persisted for at least 1 min.

## Optical mapping

Optical voltage mapping was performed using the voltage-sensitive dye, di-4-ANEPPS (15 μM; ThermoFisher Scientific, Grand Island, NY), as previously described (*Veeraraghavan et al., 2015*; *Veeraraghavan et al., 2016*; *Veeraraghavan and Poelzing, 2008*), in order to quantify longitudinal ($CV_L$) and transverse ($CV_T$) conduction velocities and anisotropy ratio (AR; the ratio of $CV_L$ to $CV_T$). During these experiments, motion was reduced using 7.5 mM 2,3-butanedione monoxime and by mechanically stabilizing the anterior epicardium against the front wall of the perfusion chamber. Ventricles were paced from a midapicobasal site on the anterior LV epicardium at basic cycle lengths of 160, 200 and 300 ms with 1 ms current pulses at 1.5 times the pacing threshold as described previously (*Veeraraghavan and Poelzing, 2008*). Preparations were excited by 510 nm light (generated from a 150 W halogen light source, passed through a 510/10 excitation filter) and fluorescent signals passed through a 610 nm longpass filter (Newport, Irvine, CA) and recorded at 1000 frames/s using a MiCAM Ultima-L CMOS camera (SciMedia, Costa Mesa, CA). Activation time was defined as the time of the maximum first derivative of the AP (*Girouard et al., 1996a*) and activation times were fitted to a parabolic surface (*Bayly et al., 1998*). Gradient vectors evaluated along this surface were averaged along the fast and slow axes of propagation (±15°) to quantify $CV_L$ and $CV_T$, respectively (*Girouard et al., 1996b*).

## Optical mapping in human iPSC-derived cardiomyocytes

Human-induced pluripotent stem-cell-derived ventricular cardiomyocytes (iPSC-CMs, Axol Biosciences, Cambridgeshire, UK) were plated on fibronectin-coated glass-bottomed chamber slides and maintained as monolayers for eight days at 37°C, 5% $CO_2$ in a humidified atmosphere in Cardiomyocyte Maintenance Medium (Axol Biosciences, Cambridgeshire, UK). Cells were loaded with the fluorescent calcium indicator Fluo-4AM in Hanks balanced salts solution using the Fluo-4 calcium imaging kit according to manufacturer's instructions (ThermoFisher Scientific, Waltham, MA). After 15 min equilibration in the microscope stage incubator (Okolab, Burlingame, CA) 20X image sequences were acquired at a rate of 400 fps on an Opterra swept-field confocal microscope (Bruker, Middleton, WI) equipped with a 510–520 nm emission filter, an Evolve Delta 512 × 512 EMCCD digital monochrome detector (Photometrics, Tucson, AZ), and a Helios 488 nm solid state laser source (Coherent, Santa Clara, CA).

## Peptide inhibitor of β1-mediated adhesion

In order to develop a selective modulator of β1-mediated adhesion, we adopted a strategy previously applied to N-cad (*Williams et al., 2000b*; *Williams et al., 2002*), and desmoglein-2 (*Schlipp et al., 2014*), wherein peptide mimetics of adhesion domains were employed as inhibitors. Thus, we identified the putative β1 adhesion domain by comparing the ectodomains of βadp1 and desmoglein-2, and developed a peptide mimetic, βadp1 (FVKILRYENEVLQLEEDERF). Additionally, we developed a scrambled control peptide, βadp1-scr (EVEQRDILEFYLLEFNVRKE), as a negative control, and a βadp1-R85D (FVKILRYENEVLQLEEDEDF), a variant peptide incorporating a substitution of the positively charged arginine at position 19 of βadp1 with a negatively charged aspartic acid. For all experiments, 20 mM stock solutions of peptides in DMSO were used.

## Statistical analysis

All data are presented as mean ± standard error unless otherwise noted, and statistical analyses were performed as previously detailed (*Veeraraghavan et al., 2015*; *Veeraraghavan et al., 2016*; *Veeraraghavan and Gourdie, 2016*). Briefly, single factor ANOVA, two factor ANOVA, or two-tailed Student's t-test with Šidák correction was applied, as appropriate, to parametric data, while Fisher's exact test was applied to non-parametric data. A $p < 0.05$ was considered statistically significant.

## Supplementary results

### Novel Na$_V$1.5 and β1 antibodies

In order to facilitate investigation of Na$_V$1.5 and β1 to ID nanodomains, we developed novel, high affinity rabbit polyclonal antibodies against C-terminal epitopes on Na$_V$1.5 and β1. Both antibodies showed strong immunofluorescent signal (green) enriched at the IDs with low background in confocal images of immunolabeled transmural sections of GP ventricular myocardium (*Figure 1—figure supplement 1A, (B)*). This immunofluorescent signal was abolished in the presence of peptides corresponding to the respective epitopes. In western immunoblots of whole cell lysates of GP ventricular myocardium run on prep gels, labeled with Na$_V$1.5 and β1 antibodies, bands were observed at 250 and 37 kDa, respectively (*Figure 1—figure supplement 1C,D* left lanes), which correspond to the molecular weights of the two proteins. Again, these signals were abolished in the presence of peptides corresponding to the respective epitopes (*Figure 1—figure supplement 1C,D* right lanes). These results demonstrate the specificity of our novel Na$_V$1.5 and β1 antibodies. Additionally, in western immunoblotting experiments, bands corresponding to β1 at 37 kDa were noted in whole cell lysates from 1610 β1OX cells and membrane lysates from the brains of WT mice, but not in whole cell lysates from 1610 Parental cells or membrane lysates from the brains of β1-null mice (*Figure 1—figure supplement 1E,F*). In dot blot experiments, our Na$_V$1.5 and β1 antibodies demonstrated selective affinity for their respective antigen peptides without cross-reactivity to each other's antigen peptides (*Figure 1—figure supplement 2*).

### Modulating β1-mediated cell-to-cell adhesion

In order to determine whether tight junctions might contribute to cell-to-cell adhesion in 1610 cells, we used confocal microscopy to assess levels of tight junction proteins: Levels of ZO-1, claudin, and occludin in 1610 cells were low or undetectable (data not shown), indicating that cell-to-cell adhesion in 1610 cells was not dependent on tight junctions. Additionally, junctional resistance in 1610β1OX cells was not significantly affected by a 50% decrease (to 0.625 mM) in extracellular Ca$^{2+}$ (109.7% of control, n = 5, p=ns), an intervention that disrupts adhesion mediated by cadherins (*Vite and Radice, 2014*), but not by β1, as reported by others (*Isom, 2002*).

## Acknowledgements

The study was supported by National Institutes of Health R01 grants awarded to RGG (RO1 HL56728-15A2; RO1 HL HL141855-01), SP (R01 HL102298-01A1; RO1 HL HL141855-01), LLI (R37NS076752), and JWS (R01 HL132236-02) and an American Heart Association Scientist Development Grant awarded to RV (16SDG29870007).

## Additional information

### Funding

| Funder | Grant reference number | Author |
|---|---|---|
| National Heart, Lung, and Blood Institute | RO1 HL56728-15A2 | Robert G Gourdie |
| American Heart Association | 16SDG29870007 | Rengasayee Veeraraghavan |
| National Heart, Lung, and Blood Institute | R01 HL102298-01A1 | Steven Poelzing |
| National Institutes of Health | R01 HL102298-01A1 | James W Smith |

| National Institutes of Health | R37NS076752 | Lori L Isom |
| National Heart, Lung, and Blood Institute | RO1 HL HL141855-01 | Robert G Gourdie Steven Poelzing |

The funders had no role in study design, data collection and interpretation, or the decision to submit the work for publication.

## Author contributions

Rengasayee Veeraraghavan, Conceptualization, Software, Formal analysis, Funding acquisition, Validation, Investigation, Visualization, Methodology, Writing—original draft, Writing—review and editing; Gregory S Hoeker, Anita Alvarez-Laviada, Formal analysis, Investigation, Methodology, Writing—review and editing; Daniel Hoagland, Investigation, Visualization; Xiaoping Wan, Formal analysis, Investigation, Methodology; D Ryan King, Jose Sanchez-Alonso, Investigation, Writing—review and editing; Chunling Chen, Resources, Methodology; Jane Jourdan, Investigation, Methodology; Lori L Isom, Resources, Funding acquisition, Methodology; Isabelle Deschenes, Julia Gorelik, Resources, Investigation, Methodology; James W Smyth, Funding acquisition, Investigation, Methodology; Steven Poelzing, Conceptualization, Resources, Funding acquisition, Investigation, Methodology, Writing—review and editing; Robert G Gourdie, Conceptualization, Resources, Supervision, Funding acquisition, Investigation, Writing—original draft, Project administration, Writing—review and editing

## Author ORCIDs

Rengasayee Veeraraghavan (ID) https://orcid.org/0000-0002-8364-2222
Robert G Gourdie (ID) http://orcid.org/0000-0001-6021-0796

## Ethics

Animal experimentation: The investigation conforms to the Guide for the Care and Use of Laboratory Animals published by the US National Institutes of Health (NIH Publication No. 85-23, revised 1996). All animal study protocols (15-130, 15-134, 12-140) were approved by the Institutional Animal Care and Use Committee at the Virginia Polytechnic University.

## Decision letter and Author response

Decision letter https://doi.org/10.7554/eLife.37610.sa1
Author response https://doi.org/10.7554/eLife.37610.sa2

## Additional files

### Supplementary files

• Transparent reporting form

### Data availability

The raw data generated in this study are available via Dryad (doi:10.5061/dryad.10351qn). The raw STORM movies are available on request from the corresponding author due to their large size.

The following dataset was generated:

| Author(s) | Year | Dataset title | Dataset URL | Database and Identifier |
|---|---|---|---|---|
| Veeraraghavan R, Hoeker G, Alvarez-Laviada A, Hoagland D, Wan X, King D, Sanchez-Alonso J, Chen C, Jourdan J, Isom L, Deschenes I, Smyth J, Gorelik | 2018 | Data from: The Adhesion Function of the Sodium Channel Beta Subunit ($\beta$1) Contributes to Cardiac Action Potential Propagation | https://dx.doi.org/10.5061/dryad.10351qn | Available at Dryad Digital Repository under a CC0 Public Domain Dedication |

J, Poelzing S,
Gourdie R

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
