## [Decision Letter]

Thank you for submitting your article "The Adhesion Function of the Sodium Channel Beta Subunit (β1) is Required for Cardiac Action Potential Propagation" for consideration by *eLife*. Your article has been reviewed by two peer reviewers, and the evaluation has been overseen by a Reviewing Editor and Richard Aldrich as the Senior Editor. The following individual involved in review of your submission has agreed to reveal her identity: Colleen Clancy (Reviewer #1).

General Assessment:

The manuscript entitled, "The Adhesion Function of the Sodium Channel Beta Subunit (β1) is Required for Cardiac Action Potential Propagation" describes combined experimental and computational studies on cardiac myocyte cell adhesion medicated by β1 protein (SCN5A), interacting with transmembrane voltage gated ion channel Na_V_1.5. β1 provides a scaffold for concentrating Na_V_1.5 in the perinexal cleft adjacent to gap junctions (GJ). It generates close cell-cell apposition within intercalated disks. A rationally designed peptide βadp1 inhibits β1 adhesion. Multiple spectroscopic, electrophysiological and modeling experiments performed by the authors provided details of this process and also its consequences for sodium channel and overall electric cardiac myocyte and tissue activities including an increased incidence of arrhythmias. The authors suggest that their findings indicate plausibility of ephatic conduction mechanism in cardiac tissue, which operates via transient changes in ion concentration in a narrow extracellular cleft rather than connexin mediated GJ connections. And extracellular β1 domain adhesion may play a major role in this mechanism, which becomes disrupted e.g. by βadp1 peptide. The authors suggest that understanding this mechanism may lead to novel ways of β1 mutation associated arrhythmias.

This is a very interesting study, which might be of substantial significance since it suggests major re-evaluation of fundamental electrical signal propagation mechanisms in the heart tissue. There are, however, a few shortcomings, which need to be addressed before the manuscript can be published.

There are certain aspects of the results that are unconvincing and/or confusing. That is to some extent inevitable given the range of techniques employed. However, in order to really make the manuscript both convincing and understandable to most readers, the authors need to improve some of the analysis of data and/or how it's described. And some of the less convincing results, in particular those in Figure 4E-F, might be better off left on the cutting room floor. Unless I misinterpreted something, these results seem to add nothing to the story.

Required Revisions:

1) First of all, it would be helpful to more clearly state the general overarching idea in the Introduction and again in the beginning of the Results section. A cartoon representation of the system/problem at hand similar to Figure 8 will be extremely useful.

2) The authors interpret results of their studies in favor of ephatic mechanisms, but it seems that it has not been confirmed directly, only through mathematical modeling and interpretation of the experiments, which suggest plausibility, but cannot prove. This should be clearly stated and some description of alternative explanations as well as possible shortcomings and alternative explanations might need to be mentioned.

3) A concise methodology section at the end of main text be very helpful.

4) Subsection “STORM-RLA Indicates Na_V_1.5 Distributes Between Two Pools within the Intercalated Disk” – connection between Figure 2 and some values described in the text needs to be made more clear. For instance: "over half of these β1 clusters localized to the perinexus" Where is it shown in Figure 2?

5) How was β1 and βadp1 docking performed?

"stabilization of the interaction via a network of intramolecular hydrogen bonds" – this does not make sense since intramolecular H bonds would typically stabilize a particular protein conformation but intermolecular H bonds would stabilize peptide – peptide interaction. How stabilities of the docked complexes were estimated?

6) "physical extent of the β1 extracellular domain […] 7-15 nm, depending on relaxation state of the domain" – that length is indeed very subjective especially considering absence of structural information in this region, necessary for accurate homology modeling.

7) Figure 2E is confusing. It is not clear why dark shading for any overlap is combined with% of clusters located in perinexus (light shading). In this case how do the clear bars can represent "no overlap" case?

8) Figure 3 parts are not clear/confusing. Panel B. There is a long completely disordered terminal fragment. Most likely, there was no homologous template structure in β3, and it was modeled de novo. It should not have been included in the homology modeling. Docked βadp1 – β1 structure is not clear as shown. A representation with two differently colored molecular surfaces and/or ribbons with a few interacting residues shown in atomic details would be much more useful.

9) Resistances in panels A and D are very different numerically. Also, there are differences in numbers in the line plots and summary bar graph in panel D. For 200 μm βadp1 values on the left graph are bigger for parental but the other way around on the right graph.

10) Figure 4F – what is the significance of low Ca^2+^ results? They do not seem to be described in the text.

11) In the Introduction in the first paragraph, it is stated that "[…] electrical excitation jumping between myocytes across extracellular clefts, not wholly unlike how activation steps discontinuously between neurons at the synaptic cleft." I'm not sure what they're trying to imply here – seems to me that ephaptic coupling *is* wholly unlike standard synaptic communication between neurons (which involves neurotransmitters and all).

12) In the Discussion it's stated that "Ephaptic coupling is known to operate in neural and other tissue". I think that ephaptic coupling is only significant in a setting when there is a high degree of synchrony and "neural mass" effects are present.

13) Except in special cases (e.g., PHP – Mauthner cell synapses), cell-cell ephaptic coupling is very weak… perhaps you could describe the typical strength of ephaptic coupling in neural tissue and its (potential) functional role.

14) In the Discussion, there's a paragraph or two of evolutionary speculation. While (or whilst!) I personally like evolutionary speculation in general, here it's probably a bit too speculative, and it really doesn't add much to the paper in my opinion. I'm actually not sure if cells in invertebrate hearts were shown to be lacking all direct electrical coupling (it would have been nice to have a citation there), but invertebrates do have gap junction proteins (inexins), and invertebrate hearts are typically neurally driven.

15) The results with the fluorescent dyes shown in Figure 4 are extremely unconvincing. First, with the 0.3 kDa dye, why the fluorescence in the ID initially *higher* than in the interstitial space? Wouldn't this most likely occur not because of diffusion, but because some binding partner is trapping the dye in the ID? Maybe I'm wrong, but that seems like the most likely explanation to me. Thus, when the peptide alters this, does it cause a change in the diffusion, or does it just disrupt the "trapping"? Although something is clearly happening with the 0.3 kDa dye, it seems premature to ascribe this effect to diffusion.

In general, diffusion is best assessed using dynamic time course techniques such as FRAP. With the snapshots presented, it's difficult to make strong inferences.

The other issue here is that the peptide causes no change at all to the distributions of the 3kDa and 10kDa dyes. However, low [Ca] does, although this result is shown in the figure yet not described at all in the text. Overall these results are confusing and difficult to interpret, and the authors' conclusions about diffusion barriers is not compelling.

16) The results shown in Figure 3, with a technology called ECIS, show convincing trends, but what exactly is being measured in these experiments is not at all clear. What's confusing is that the text refers to measurements of "intercellular junctional resistance." Does this mean resistance through the gap junctions connecting adjacent cells? In the context of cardiac electrophysiology, that's what this term usually means. But the trends seen in the Results seem to indicate that something different is being measured, which makes this part confusing.

For instance, I one would expect the gap junctional resistance to be much *lower* in the 1610 β10X cells than in the 1610 parental cells. And then I would expect the peptide to *increase* this resistance in the β10X cells. But instead the exact opposite trends are observed.

So, what is the ECIS technique actually measuring? It must be some form of impedance from one electrode to another electrode in the extracellular space? But can this resistance be directly related to the cable properties that help to determine conduction? This section requires much clearer explanations of what is being measured and what the results mean in terms of physiological parameters.

17) It is convincing in Figure 7 that the peptide widens the QRS complex and slows propagation. But why should this be anisotropic? If the IDs are located at the cell ends, disruption of these ephapses would be expected to slow both longitudinal and transverse condition, with no a priori reason to think there should be a change in the anisotropy ratio. The manuscript repeatedly mentions the 2015 Pflugers Archiv paper for justification for this, but without offering any explanation. Readers of the journal shouldn't have to also examine prior studies in order to understand these results.

18) The results in Figure 6 are interesting, but the interpretation is not totally clear. If the peptide was just increasing the width of the ID between myocytes, this would affect the ephaptic conduction, without necessarily decreasing the amount of Na current at the ephapse. So it seem like the β subunit is also keeping the channels in that location, besides controlling the width of the gap.

19) The colocalization analyses shown in Figure 2 are quite confusing and not well-described. The text requires the reader to slog through lots and lots of numbers, without a good sense of the big picture implications. And when one examines the figure, it's not at all clear how "incidence of overlap" in Figure 2E should be different from "degree of overlap" in Figure 2F. The difference between these measured is only mentioned in passing. Sometimes 2E and 2F are consistent with each other, other times not. This is all quite confusing.

---

## [Author Response]

[…] There are certain aspects of the results that are unconvincing and/or confusing. That is to some extent inevitable given the range of techniques employed. However, in order to really make the manuscript both convincing and understandable to most readers, the authors need to improve some of the analysis of data and/or how it's described. And some of the less convincing results, in particular those in Figure 4E-F, might be better off left on the cutting room floor. Unless I misinterpreted something, these results seem to add nothing to the story.

We thank the editor, and reviewers for their thoughtful evaluation of our study, and have significantly revised the manuscript in response. Foremost, we appreciate the key issues identified by the reviewers which cofound interpretation of our dye perfusion studies. Given that these results are not essential to our primary conclusions, we have removed the dye perfusion experiments from the manuscript. Additionally, we have significantly revised Figure 2, and the text describing the STORM results with the goal of making them easier to comprehend.

Required Revisions:1) First of all, it would be helpful to more clearly state the general overarching idea in the Introduction and again in the beginning of the Results section. A cartoon representation of the system/problem at hand similar to Figure 8 will be extremely useful.

We thank the reviewer for this helpful suggestion and have revised the closing paragraph of the Introduction, and added an opening paragraph to the Results section outlining the overarching ideas tested in our study. Likewise, we have revised Figure 1B, adding a cartoon illustrating the problem at hand.

2) The authors interpret results of their studies in favor of ephatic mechanisms, but it seems that it has not been confirmed directly, only through mathematical modeling and interpretation of the experiments, which suggest plausibility, but cannot prove. This should be clearly stated and some description of alternative explanations as well as possible shortcomings and alternative explanations might need to be mentioned.

The reviewer’s point is well-taken. We have added text to the last paragraph of the Discussion softening claims regarding ephaptic coupling.

3) A concise methodology section at the end of main text be very helpful.

We have added a brief account of the methods to the paper with a fuller description included in the supplement.

4) Subsection “STORM-RLA Indicates Na_V_1.5 Distributes Between Two Pools within the Intercalated Disk” – connection between Figure 2 and some values described in the text needs to be made more clear. For instance: "over half of these β1 clusters localized to the perinexus" Where is it shown in Figure 2?

We have now revised Figure 2, and significantly rewritten the section describing the STORM results (subsection “STORM-RLA Indicates Na_V_1.5 Distributes Between Two Pools within the Intercalated Disk”), with the aim of making it simpler, and clearer. We hope that these revisions adequately clarify the inferences drawn from this complex dataset.

5) How was β1 and βadp1 docking performed?"stabilization of the interaction via a network of intramolecular hydrogen bonds" – this does not make sense since intramolecular H bonds would typically stabilize a particular protein conformation but intermolecular H bonds would stabilize peptide – peptide interaction. How stabilities of the docked complexes were estimated?

We have rewritten the section on molecular modeling noting the software used, and reworded the binding description – intramolecular bonds stabilize the binding pose, which lowers binding energy. Electrostatic/hydrophobic interactions are peptide-protein. This has now been clarified in the manuscript and we have added the free energy of binding from MMGBSA refinement of poses to the text.

6) "physical extent of the β1 extracellular domain[…] 7-15 nm, depending on relaxation state of the domain" – that length is indeed very subjective especially considering absence of structural information in this region, necessary for accurate homology modeling.

Structural information from the β3 crystal structure approximate the size of that extracellular loop to be ~5nm (DOI: 10.1098/rsob.140192, Figure 3), and β1 is slightly larger. A homology model based on a 4.0 Å cryo-EM of the Na_V_1.4-β1 complex from the electric eel (Figure 3B) shows that there is a relatively unorganized region linking the Ig loop and the transmembrane domain, fully capable of extending the length between the membrane and the furthest regions of the protein by a few nm. These results support the possibility that two apposed β1 subunits can span a 10-15 nm extracellular cleft. We have added this improved visualization of our β1 molecular model to Figure 3B, and revised the text (subsection “βadp1 – A Rationally Designed Inhibitor of β1-mediated Adhesion”, second paragraph) to better clarify the modeling approach as noted above.

7) Figure 2E is confusing. It is not clear why dark shading for any overlap is combined with% of clusters located in perinexus (light shading). In this case how do the clear bars can represent "no overlap" case?

We appreciate the reviewers pointing out this confusing labeling in this figure, and have now revised it. In Figure 2A (previously 2E), each bar is now divided into 3 segments: overlap (colored), adjacent (perinexal) (shaded), distant (clear). We hope that this resolves the issues stemming from our formerly confusing labeling.

8) Figure 3 parts are not clear/confusing. Panel B. There is a long completely disordered terminal fragment. Most likely, there was no homologous template structure in β3, and it was modeled de novo. It should not have been included in the homology modeling. Docked βadp1 – β1 structure is not clear as shown. A representation with two differently colored molecular surfaces and/or ribbons with a few interacting residues shown in atomic details would be much more useful.

We thank the reviewer for these very helpful suggestions. The docking studies referenced in the text were performed with an updated homology model that incorporated structural data from a 4.0 Å cryo-EM of the Na_V_1.4 channel in complex with β1. The binding surface depicted was unchanged upon updating the model, but Figure 3B now more accurately reflects the structure between the IG loop and the transmembrane domain. The updated view in Figure 3C removes the surface rendering of the β1 Ig loop and shows atomic details of βadp1 binding (ball and stick) and β1 surface (cartoon with interacting residues as stick).

9) Resistances in panels A and D are very different numerically. Also, there are differences in numbers in the line plots and summary bar graph in panel D. For 200 μm βadp1 values on the left graph are bigger for parental but the other way around on the right graph.

The initial experiments comparing 1610β1WT and 1610 Parental cells were carried out with 4 x 10^4^ cells seeded per well, whereas the experiments with βadp1 were conducted with 1 x 10^4^ cells per well. Hence, the difference in the absolute values measured. Additionally, the line graphs (Figure 3D, left) present representative results from a single ECIS experiment whereas the bar graph (Figure 3D, right) presents summary data from 5 experiments, each with 2 technical replicates per cell type and treatment condition. These details are now better clarified in the figure legend.

10) Figure 4F – what is the significance of low Ca^2+^ results? They do not seem to be described in the text.

Based on the reviewers’ feedback, we have removed the dye perfusion studies from the manuscript.

11) In the Introduction in the first paragraph, it is stated that "[…] electrical excitation jumping between myocytes across extracellular clefts, not wholly unlike how activation steps discontinuously between neurons at the synaptic cleft." I'm not sure what they're trying to imply here – seems to me that ephaptic coupling is wholly unlike standard synaptic communication between neurons (which involves neurotransmitters and all).

This sentence was meant to draw a distinction between GJ coupling, which occurs via cytoplasmic continuity, and synaptic and ephaptic mechanisms, both of which work across narrow extracellular clefts, albeit in different ways. In order to avoid confusion, we have removed this sentence from the manuscript.

12) In the Discussion it's stated that "Ephaptic coupling is known to operate in neural and other tissue". I think that ephaptic coupling is only significant in a setting when there is a high degree of synchrony and "neural mass" effects are present.13) Except in special cases (e.g., PHP – Mauthner cell synapses), cell-cell ephaptic coupling is very weak. Perhaps you could describe the typical strength of ephaptic coupling in neural tissue and its (potential) functional role.

We agree with the reviewer on the relative role of ephaptic coupling in neural tissues, vis-à-vis other coupling mechanisms. We were simply making the point that prior experimental observations of ephaptic coupling had been made in neural and retinal tissues, unlike the case with the heart. To make this clearer, we have softened the wording of this statement in the second paragraph of the Discussion.

14) In the Discussion, there's a paragraph or two of evolutionary speculation. While (or whilst!) I personally like evolutionary speculation in general, here it's probably a bit too speculative, and it really doesn't add much to the paper in my opinion. I'm actually not sure if cells in invertebrate hearts were shown to be lacking all direct electrical coupling (it would have been nice to have a citation there), but invertebrates do have gap junction proteins (inexins), and invertebrate hearts are typically neurally driven.

We agree with the reviewer that this discussion distracts from the main points of the manuscript; therefore, we have removed this section.

15) The results with the fluorescent dyes shown in Figure 4 are extremely unconvincing. First, with the 0.3 kDa dye, why the fluorescence in the ID initially higher than in the interstitial space? Wouldn't this most likely occur not because of diffusion, but because some binding partner is trapping the dye in the ID? Maybe I'm wrong, but that seems like the most likely explanation to me. Thus, when the peptide alters this, does it cause a change in the diffusion, or does it just disrupt the "trapping"? Although something is clearly happening with the 0.3 kDa dye, it seems premature to ascribe this effect to diffusion.In general, diffusion is best assessed using dynamic time course techniques such as FRAP. With the snapshots presented, it's difficult to make strong inferences.The other issue here is that the peptide causes no change at all to the distributions of the 3kDa and 10kDa dyes. However, low [Ca] does, although this result is shown in the figure yet not described at all in the text. Overall these results are confusing and difficult to interpret, and the authors' conclusions about diffusion barriers is not compelling.

We thank the reviewer for pointing out some key issues that cofound interpretation of our dye perfusion studies. Given that these results are not essential to our primary conclusions, we have removed the dye perfusion experiments from the manuscript.

16) The results shown in Figure 3, with a technology called ECIS, show convincing trends, but what exactly is being measured in these experiments is not at all clear. What's confusing is that the text refers to measurements of "intercellular junctional resistance." Does this mean resistance through the gap junctions connecting adjacent cells? […] This section requires much clearer explanations of what is being measured and what the results mean in terms of physiological parameters.

The ECIS technique does indeed rely on measurement of impedance between electrodes placed across a monolayer of cells. Specifically, this technique measures the electrical impedance offered by a monolayer of cells to current flow between electrodes located above, and below the monolayer, and previous studies demonstrate that the resistance of the extracellular cleft at cell-cell junctions (junctional resistance) is well-reflected by measurements at low frequencies (62.5-4000 Hz) (Williams, Williams and Doherty, 2002; Schlipp et al., 2014). This is now clearly stated in the manuscript (subsection “βadp1 – A Rationally Designed Inhibitor of β1-mediated Adhesion”, first paragraph). Since the technique measures impedance between two extracellular compartments (above, and below the cells), it is incapable of assessing gap junction resistance connecting the intracellular compartments of adjacent cells.

17) It is convincing in Figure 7 that the peptide widens the QRS complex and slows propagation. But why should this be anisotropic? If the IDs are located at the cell ends, disruption of these ephapses would be expected to slow both longitudinal and transverse condition, with no a priori reason to think there should be a change in the anisotropy ratio. The manuscript repeatedly mentions the 2015 Pflugers Archiv paper for justification for this, but without offering any explanation. Readers of the journal shouldn't have to also examine prior studies in order to understand these results.

We thank the reviewer for raising this excellent question. We reason that the anisotropic effects of βadp1 treatment reflect the fact that a wavefront traveling transverse to the fiber axis would encounter more cell-cell junctions per unit distance than one traveling longitudinally. Thus, any compromise in cell-cell coupling via perinexal nanodomains would have a greater impact on transverse conduction, than on longitudinal conduction. We have now added this explanation to the sixth paragraph of the Discussion.

18) The results in Figure 6 are interesting, but the interpretation is not totally clear. If the peptide was just increasing the width of the ID between myocytes, this would affect the ephaptic conduction, without necessarily decreasing the amount of Na current at the ephapse. So it seem like the β subunit is also keeping the channels in that location, besides controlling the width of the gap.

We do indeed agree with the reviewer on this point. The β1 subunit likely performs multiple functions in the generation, and maintenance of ephaptic nanodomains.

19) The colocalization analyses shown in Figure 2 are quite confusing and not well-described. The text requires the reader to slog through lots and lots of numbers, without a good sense of the big picture implications. And when one examines the figure, it's not at all clear how "incidence of overlap" in Figure 2E should be different from "degree of overlap" in Figure 2F. The difference between these measured is only mentioned in passing. Sometimes 2E and 2F are consistent with each other, other times not. This is all quite confusing.

We have now revised Figure 2 and significantly rewritten the section describing the STORM results (subsection “STORM-RLA Indicates Na_V_1.5 Distributes Between Two Pools within the Intercalated Disk”), with the aim of making it simpler, and clearer. Briefly, ‘Incidence of overlap’ referred to the percentage of clusters which demonstrated any overlap at all, while ‘degree of overlap’ referred to the fractional volume of participating clusters involved in the overlap. To avoid potential confusion, we have now relabeled the incidence of overlap as “relative localization overview” and noted that the solid, shaded, and uncolored bars refer respectively to “any overlap”, “adjacent”, and “distant” situations. We hope that these revisions to the figure, and the text have mitigated the issues noted by the reviewer.